**EMBO** *reports*

# Histone modifications and Sp1 promote GPR160 expression in bone cancer pain within rodent models

Chengfei Xu [1,2,8], Yahui Wang [1,8], Chaobo Ni[1], Miao Xu[1], Chengyu Yin[3], Qiuli He[1], Bing Ma[2], Jie Fu[1], Baoxia Zhao[1], Liping Chen[1], Tong Zhi[1], Shirong Wei[1], Liang Cheng[2], Hui Xu[4], Jiajun Xiao[5], Lei Yang[1], Qingqing Xu[1], Jiao Kuang[1], Boyi Liu[6], Qinghe Zhou [1], Xuewu Lin [7✉], Ming Yao [1✉] & Huadong Ni [1✉]

## Abstract

**Bone cancer pain (BCP) affects ~70% of patients in advanced stages, primarily due to bone metastasis, presenting a substantial therapeutic challenge. Here, we profile orphan G protein-coupled receptors in the dorsal root ganglia (DRG) following tumor infiltration, and observe a notable increase in GPR160 expression. Elevated *Gpr160* mRNA and protein levels persist from postoperative day 6 for over 18 days in the affected DRG, predominantly in small-diameter C-fiber type neurons specific to the tibia. Targeted interventions, including DRG microinjection of siRNA or AAV delivery, mitigate mechanical allodynia, cold, and heat hyperalgesia induced by the tumor. Tumor infiltration increases DRG neuron excitability in wild-type mice, but not in *Gpr160* gene knockout mice. Tumor infiltration results in reduced H3K27me3 and increased H3K27ac modifications, enhanced binding of the transcription activator Sp1 to the *Gpr160* gene promoter region, and induction of GPR160 expression. Modulating histone-modifying enzymes effectively alleviated pain behavior. Our study delineates a novel mechanism wherein elevated Sp1 levels facilitate *Gpr160* gene transcription in nociceptive DRG neurons during BCP in rodents.**

**Keywords** Bone Cancer Pain; Histone Modification; GPR160; Dorsal Root Ganglia; Peripheral Sensitization
**Subject Categories** Cancer; Chromatin, Transcription & Genomics; Neuroscience

## Introduction

Bone cancer pain (BCP) is the most common and refractory manifestation of pain in cancer patients, particularly among those afflicted with metastatic breast, lung, and prostate malignancies (Mantyh, 2013). The neurobiological underpinnings of cancer-induced pain are intricate (Goblirsch et al, 2006). Presently employed therapeutic approaches involving opioids and nonsteroidal anti-inflammatory drugs remain notably unsatisfactory due to their limited efficacy and the associated risk of addiction (Arthur and Bruera, 2022; Hunter, 2018). A mounting body of evidence has surfaced, indicating the potential involvement of epigenetic mechanisms in various pain syndromes (Ni et al, 2024; Xie et al, 2023). These epigenetic alterations encompass histone modifications, DNA methylation, and noncoding RNAs, all of which undergo modifications in response to peripheral noxious stimuli. These changes affect the expression of genes related to pain and contribute to the development of pain (Descalzi et al, 2015).

Dorsal root ganglia (DRG) house primary somatosensory neurons that contribute to relaying peripheral nociceptive signals to central terminals (Chen et al, 2019; Sharma et al, 2024). Epigenetic modifications in the DRG are essential in the initiation, progression, and maintenance of pain (Zhang et al, 2021a). Notably, conditions encompassing spinal nerve ligation (SNL) and chronic constriction injury of the sciatic nerve (CCI) have been shown to elevate the expression of DNMT1 and DNMT3a within injured DRG (Sun et al, 2019; Zhao et al, 2017). Given that the initial pathological alterations in the DRG are important elements in initiating chronic pain, targeting epigenetic patterns within the DRG holds promise for effectively averting its development (Li et al, 2019). Hence, investigating the involvement of DRG epigenetics in nociceptive processing holds significant potential for advancing chronic pain management.

G protein-coupled receptors (GPCRs), distinguished by their seven-transmembrane architecture, hold paramount significance in intracellular signal transduction, being implicated in a myriad of physiological and

[1]Department of Anesthesiology and Pain Research Center, The Affiliated Hospital of Jiaxing University, 1882 Zhonghuan South Road, 314001 Jiaxing, China. [2]Department of Anesthesiology, The Third People's Hospital of Bengbu, 38 Shengli Middle Road, 233000 Bengbu, China. [3]Department of Integrated Traditional Chinese and Western Medicine, Union Hospital, Tongji Medical College, Huazhong University of Science and Technology, 430022 Wuhan, China. [4]Department of Anesthesiology, The First People's Hospital of Bengbu, 233000 Bengbu, China. [5]Bengbu Hospital of Traditional Chinese Medicine, 4339 Huai-Shang Road, 233000 Bengbu, China. [6]Department of Neurobiology and Acupuncture Research, Key Laboratory of Acupuncture and Neurology of Zhejiang Province, The Third Clinical Medical College, Zhejiang Chinese Medical University, 310053 Hangzhou, China. [7]Department of Pain Medicine, The First Affiliated Hospital of Bengbu Medical University, 233000 Bengbu, China. [8]These authors contributed equally: Chengfei Xu, Yahui Wang. ✉E-mail: linxuewu@bbmc.edu.cn; jxyaoming@zjxu.edu.cn; huadongni@zjxu.edu.cn

pathological processes (Hauser et al, 2018; Schou et al, 2015). Notably, ~34% of currently marketed pharmaceuticals exert their effects directly through GPCRs (Santos et al, 2017). Within the domain of pain medicine, GPCRs emerge as pivotal therapeutic targets. Orphan GPCRs (oGPCRs), a subset of receptors devoid of established endogenous ligands and with partially elucidated physiological functions, have garnered attention. Among these, GPR34, GPR84, GPR151, and GPR177 have been identified in the DRG or spinal cord, playing pivotal roles in the development and maintenance of neuropathic pain (Jiang et al, 2021; Nicol et al, 2015; Sayo et al, 2019; Xia et al, 2021; Xie et al, 2022). However, the expression and role of oGPCRs within the DRG in the context of BCP remain unclear. In our quest for novel oGPCRs involved in the regulation of BCP, we conducted an analysis of mRNA microarray data obtained following tumor infiltration-induced BCP. Intriguingly, orphan GPR160 was significantly upregulated in DRGs following tumor infiltration. In the neuropathic pain model, GPR160 exhibited upregulation in spinal neurons and was manifested to be involved in maintaining neuropathic pain generated by SNI/CCI (Yosten et al, 2020). Nevertheless, the extent of GPR160's involvement in BCP remains a subject of ongoing debate.

In this investigation, we elucidate that GPR160 exhibits predominant expression within small-diameter DRG neurons and is essential in BCP pathogenesis. The modulation of histone modifications serves to enhance the binding capacity of the transcription factor Sp1 to the *Gpr160* gene promoter, consequently increasing GPR160 expression following tumor infiltration. Furthermore, our findings underscore the profound impact of GPR160 expression levels on the development and maintenance of BCP. This mechanistic insight underscores the potential therapeutic efficacy of targeting the GPR160 for BCP treatment.

# Results

## GPR160 is persistently upregulated in DRG neurons after tumor infiltration

We established a rat model of BCP by injecting Walker 256 cells into the tibia as described (Ni et al, 2023). Control rat received heat-inactivated cells injections. Following intra-tibial infiltration of Walker 256 cells, rats exhibited persistent mechanical allodynia, thermal hyperalgesia, and cold allodynia by postoperative day (POD) 6 in the ipsilateral hind paw, extending for a minimum of 18 days (Fig. EV1A–C). Given the consistent onset of pain-like behaviors by POD 12, this time point was chosen for subsequent experiments. CatWalk gait analysis, recognized for its value, has demonstrated associations with mechanical allodynia and spontaneous pain in various chronic pain models (Hu et al, 2019). In Fig. EV1D–H, we noted a significant reduction in max contact area, max contact mean intensity, and single stance percentages, alongside a notable increase in swing percentage on the plantar aspect of BCP rats. These observations suggest a propensity among BCP rats to minimize weight-bearing on the affected hind limb during ambulation. In the open field test (Fig. EV1I–K), BCP rats demonstrated an anxiety-like behavior, as evidenced by a significant reduction in both the time spent and the distance moved exploring the central area. Hematoxylin–eosin staining (Fig. EV1L) revealed concurrent tumor growth and bone destruction, accompanied by medullary bone loss. Additionally, we utilized computed tomography (CT) with three-dimensional reconstruction to visualize bone destruction (Fig. EV1M). These findings substantiate the effective establishment of the BCP model in rats.

In our quest for novel oGPCR genes implicated in the pathogenesis of BCP, we performed mRNA microarray of the ipsilateral DRG on POD 12. The resulting cluster heatmap and volcano plot vividly depicted mRNA expression variations between BCP and sham-operated control samples (Fig. 1A,B). Among these differentially expressed oGPCR genes identified (fold change [FC] >2), 8 genes exhibited upregulation in BCP rats as compared to the controls. Subsequently, we examined the expression profiles of the eight most significantly upregulated oGPCR mRNAs (*Gpr18*, *Gpr65*, *Gpr183*, *Gpr160*, *Gpr84*, *Gpr31*, *Mrgprx3*, and *Gpr132*, as enumerated in Fig. 1C) within the DRG of BCP rats. Notably, GPR65, among these genes, is known to modulate hypersensitivity in a rat model of BCP. Our focus turned towards GPR160, a hitherto unexplored gene in BCP yet implicated in neuropathic pain conditions (Yosten et al, 2020). Thus, we embarked on investigating the potential involvement of GPR160 in mediating the pain response within the BCP model. Our observations revealed a time-dependent upregulation of *Gpr160* mRNA and GPR160 protein expression within the ipsilateral DRG due to BCP (Fig. 1D,E), while the contralateral DRG displayed no such alterations (Fig. EV1N).

Immunofluorescence (IF) analysis further revealed a few GPR160 neurons scattered in the DRG of both naive and sham-operated rats (Fig. 1G,I). However, a substantial increase in GPR160-positive neurons was evident in BCP rats on POD 12 (Fig. 1J). The specificity of the antibody was validated using a blocking peptide corresponding to the GPR160 antigen sequence and knockout mice (Fig. 1H; Appendix Fig. S1D). Statistical analysis demonstrated a significant increase in GPR160 cells on POD 12 (Fig. 1F). Double immunofluorescence elucidated complete co-localization of GPR160 with the neuronal marker NeuN (Fig. 1K–O) but not with the satellite glial cell marker glutamine synthetase (GS) (Fig. 1P–T), suggesting the neuronal expression of GPR160 within the DRG. These findings strongly implicate a potential role of GPR160 expressed in peripheral sensory neurons in the context of BCP.

## GPR160 is expressed in tibia-specific DRG neurons

To ascertain the source of primary afferent innervation within the tibia, we conducted IF analysis to elucidate the expression profile of GPR160 in the DRG neurons that innervate the tibia. These neurons were retrogradely labeled by DiI injection into the tibia cavity. The results of IF demonstrated the presence of GPR160 expression in 27.7% of DiI-labeled sensory neurons innervating the tibia, regardless of their cell size (Fig. 2A–E). Notably, the IF signal of GPR160 was co-localized with immunostaining for the neuronal-specific nuclear protein NeuN.

To further clarify the GPR160 expression pattern within tibia-specific sensory DRG neurons, a triple staining approach targeting DiI, GPR160, and distinct cellular markers was employed. Within the cohort of DiI-labeled sensory neurons innervating the tibia, GPR160 expression was discerned in both small-diameter (isolectin B4 [IB4], calcitonin gene-related peptide [CGRP]) and large-diameter (neurofilament-200 [NF200]) neurons. Specifically, 31.3% of GPR160-positive neurons co-expressed IB4, a characteristic marker for non-peptidergic C-fibers (Fig. 2F–J), while 49.8% co-expressed CGRP, a marker for peptidergic C-fiber and a subset of Aδ-fibers (McCoy et al, 2013) (Fig. 2K–O). Additionally, 33.5% of GPR160-positive neurons co-expressed NF200, indicative of myelinated A-fiber and a portion of Aδ fibers (Fig. 2P–T). These findings indicate that a specific subset of sensory neurons innervating the tibia, including nociceptors, expresses GPR160. This establishes the anatomical basis for the modulation of BCP through neuronal GPR160.

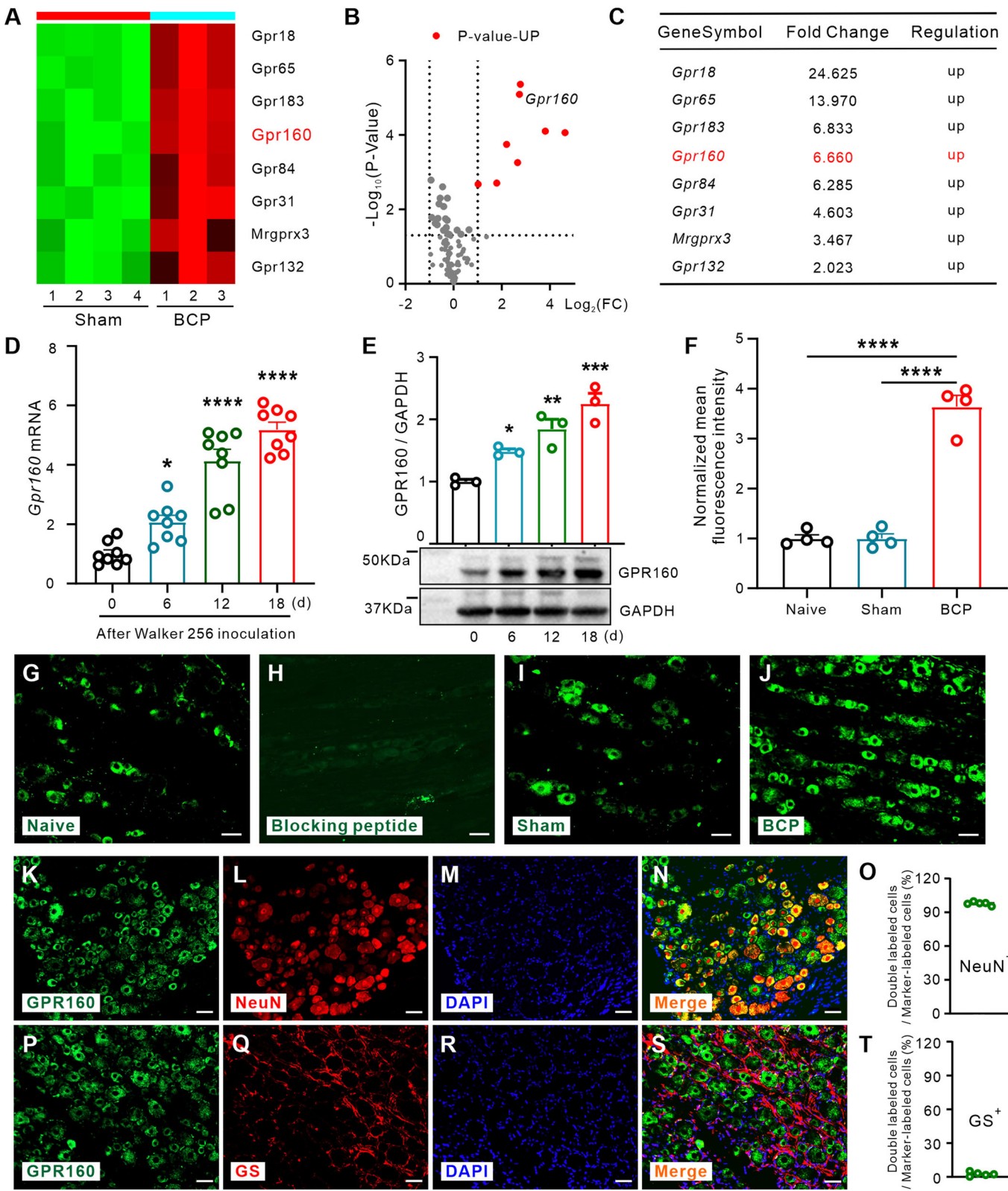

**Figure 1. GPR160 expression is upregulated in the dorsal root ganglia (DRG) after BCP.**

(A) The heatmap displays the top 8 upregulated oGPCR genes in the DRG following BCP. (B) Volcano plot illustrates gene expression profiles comparing the BCP group to the Sham group, with upregulated genes represented by red spots. $n = 3/4$ rats, Student's unpaired $t$ test. (C) The fold change in upregulated oGPCR genes, assessed 12 days post-BCP compared with sham-treated rats, was considered significant at a threshold of $\geq$ two-fold change. (D) The time course of *Gpr160* mRNA expression in the ipsilateral DRG of rats following tumor infiltration. Data are mean ± SEM of biological replicates $n = 8$ rats/group. One-way ANOVA with repeated measures followed by post hoc Tukey test. $*P = 0.0244$, $****P < 0.0001$ versus 0 d group. (E) The time course of GPR160 protein expression in the ipsilateral DRG of rats following tumor infiltration. Data are mean ± SEM of biological replicates $n = 3$ rats/group. One-way ANOVA with repeated measures followed by post hoc Tukey test. $*P = 0.0499$, $**P = 0.0045$, $***P = 0.0003$ versus 0 d group. (F) Statistical analysis shows GPR160 intensity in the ipsilateral DRG of naive, Sham and BCP rats. One-way ANOVA with repeated measures followed by post hoc Tukey test, Data are mean ± SEM of biological replicates $n = 4$ rats, $****P < 0.0001$ versus naive or Sham group. (G–J) GPR160 expression was assessed in the DRG of naive (G), sham (I), and BCP rats (J). Specificity control for the GPR160 antibody was performed through pre-incubation with an excess of the GPR160 blocking peptide (H). Scale bar, 50 μm. (K–T) Dual immunostaining for GPR160/NeuN (N) and GPR160/GS (O) was conducted in the DRG 12 days after BCP. Quantitative analysis was performed to determine the percentage of overlap. $n = 5$ rats, Scale bar, 50 μm. Source data are available online for this figure.

## Blocking elevated GPR160 in the injured DRG mitigates bone cancer pain

Does upregulated GPR160 in the injured DRG play a role in BCP hypersensitivity? To explore this, we initially assessed the impact of suppressing the elevated levels of GPR160 in the DRG on the development of pain hypersensitivity induced by tumor infiltration. Microinjections of *Gpr160* siRNA (siGpr160) and its corresponding scrambled siRNA (scram) were administered into the ipsilateral DRG on POD 3 and POD 10. The efficacy of *Gpr160* siRNA was validated through its ability to downregulate GPR160 in the ipsilateral DRG on POD 7 and 14, respectively (Fig. EV2A–D). Microinjecting *Gpr160* siRNA on POD 3, a stage prior to the manifestation of BCP hyperalgesia, resulted in a reduction of mechanical allodynia and heat hyperalgesia, although cold allodynia remained unaffected when compared to BCP rats treated with scrambled siRNA (Fig. EV2E–G). Notably, no alterations were observed in the basal levels of mechanical, heat, or cold allodynia on the ipsilateral side of sham rats subsequent to the microinjection of either siRNA into the DRG. Furthermore, we explored the involvement of DRG GPR160 in the maintenance of BCP by administering siRNA via microinjection into the ipsilateral DRG on POD 10. Consistently, a notable mitigation of mechanical allodynia, heat hyperalgesia, and cold allodynia was observed on days 12 and 14 following tumor infiltration on the ipsilateral side in rats treated with *Gpr160* siRNA, as compared to those treated with scrambled siRNA (Fig. EV2H–J). Importantly, the basal responses on the ipsilateral side were not impacted by either siRNA treatment (Fig. EV2E–J) and locomotor functions remained unchanged (Appendix Table S1).

To further ascertain the GPR160 function in the DRG concerning the development and maintenance of BCP, we administered microinjections of adeno-associated virus serotype 2/9 (AAV2/9) expressing *Gpr160* shRNA (AAV2/9-*Gpr160* shRNA) into the ipsilateral DRGs either 28 or 16 days prior to BCP or sham surgery (Fig. 3A,B), considering that AAV2/9 requires 4 weeks to achieve optimal expression. AAV2/9 carrying scrambled shRNA (AAV2/9-Scr shRNA) served as the control. As anticipated, histological examination of microinjected DRGs stained with hematoxylin/eosin affirmed their structural integrity, displaying normal neuronal morphology and an absence of evident immune cell infiltration (Fig. EV3A). The microinjection of AAV2/9-*Gpr160*-Gfp exhibited a marked GFP expression in ipsilateral DRG but not spinal cord neurons, providing definitive evidence of successful AAV2/9 delivery (Fig. EV3B,C). As anticipated, the GPR160 protein levels on POD 18 were notably diminished in the ipsilateral DRG of rats subjected to AAV2/9-*Gpr160* shRNA microinjections conducted 28 and 16 days prior to tumor cell injection, in comparison to those microinjected with AAV2/9-Scr shRNA (Fig. EV3D–G). Subsequently, we evaluated the

behavioral responses following AAV2/9 microinjections during the development of nociceptive hypersensitivity induced by BCP. Similar to the observation above, the microinjection of AAV2/9-*Gpr160* shRNA 28 days prior to surgery notably attenuated the BCP-induced enhancements in PWT and the BCP-induced reductions in PWL to heat and cold stimuli (Fig. 3C–E). All rats subjected to microinjections displayed unaltered locomotor function (Appendix Table S1). To determine the further contribution of heightened DRG GPR160 in the maintenance of BCP, we conducted microinjections of AAV2/9-*Gpr160* shRNA 16 days before the surgical procedure. Subsequently, post-surgery, both AAV2/9-*Gpr160* shRNA-microinjected and AAV2/9-Scr shRNA-microinjected rats exhibited complete development of mechanical allodynia, heat hyperalgesia and cold allodynia (Fig. 3F-H). Nonetheless, solely mechanical allodynia and enhanced heat nociceptive sensitivities exhibited a reduction in rats microinjected with AAV2/9-*Gpr160* shRNA on POD 12 and 18 following BCP surgery, with no significant impact observed on responses to cold stimuli (Fig. 3F–H). Collectively, these results robustly advocate that the upregulation of GPR160 within the injured DRG is imperative for both the development and maintenance of nociceptive hypersensitivity induced by BCP.

## DRG GPR160 overexpression contributes to pain hypersensitivity

Furthermore, we investigated whether the increased DRG GPR160 alone was adequate to induce nociceptive hypersensitivity due to tumor infiltration. To explore this, we administered microinjections of AAV2/9 carrying the full-length GPR160 (AAV2/9-*Gpr160*) into the unilateral DRG of adult naive rats (Fig. EV4A), employing AAV2/9-*Gfp* as the control. As anticipated, a notable increase in GPR160 protein levels and *Gpr160* mRNA was observed 5 weeks post-administration of AAV2/9-*Gpr160*, compared to the AAV2/9-Gfp group (Fig. EV4B–D). Rats injected with AAV2/9-*Gpr160* (but not AAV2/9-*Gfp*) exhibited a significant decrease in PWT when subjected to mechanical stimulation, as well as a decrease in PWL when exposed to heat and cold stimuli on the ipsilateral side (Fig. 3I–K). These reductions were evident at the 5-week post-injection mark and endured for a minimum of 7 weeks. The microinjection of LV-*Gpr160*, which expresses the full-length GPR160, into the ipsilateral DRG of naive rats resulted in similar outcomes (Fig. EV4E–G). Additionally, DRG microinjection of AAV2/9-*Gpr160* induced nociceptive hypersensitivity independent of evoked stimulation, as evidenced by gait analysis conducted during the 5th week post-microinjection (Fig. EV4H–L). Beyond evoked nociceptive hypersensitivity, the administration of AAV2/9-*Gpr160* into the DRG also elicited

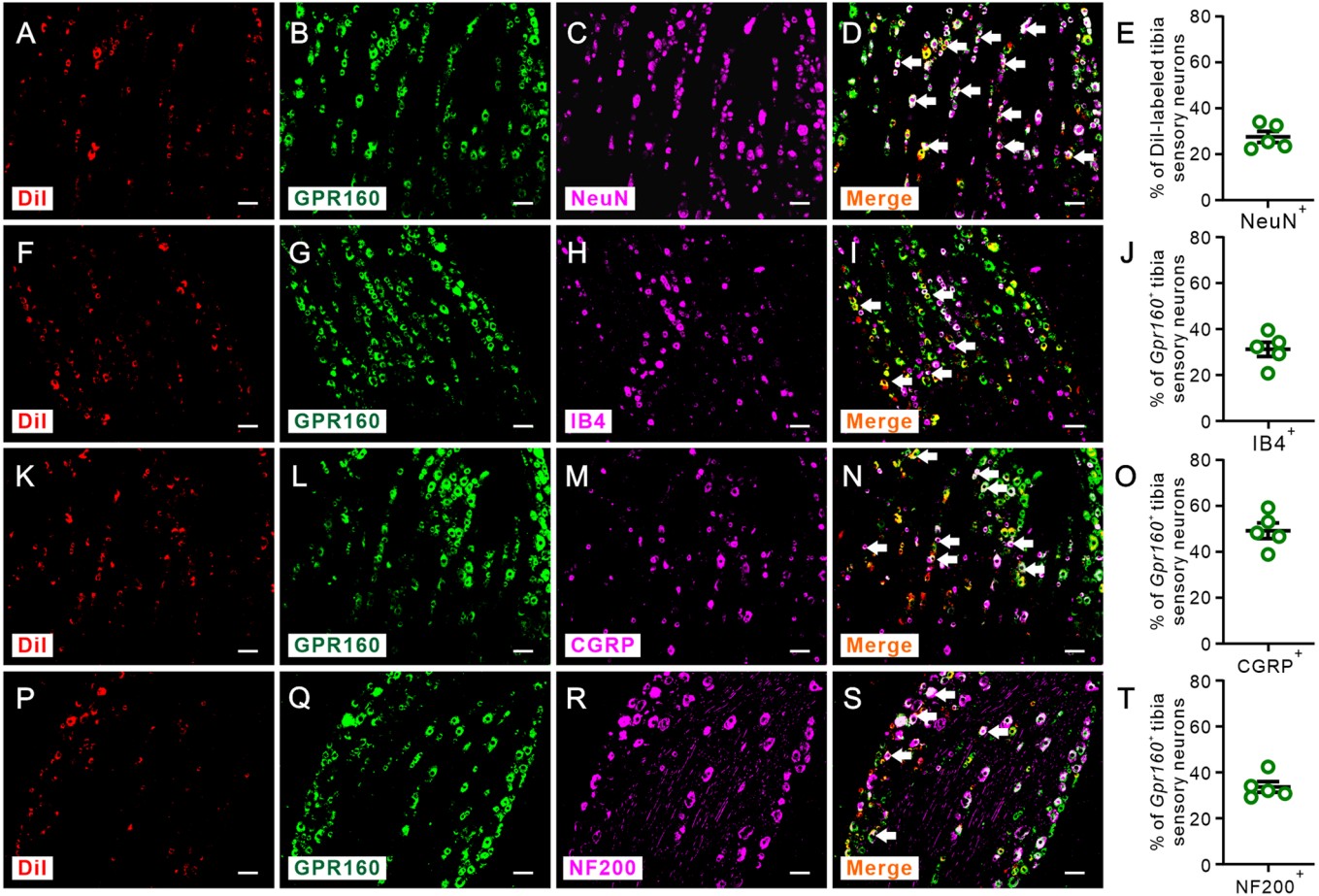

**Figure 2. Analysis of GPR160 expression in rat tibia sensory neurons.**

DRG neurons innervating the tibia were retrogradely labeled by DiI injection into the tibia cavity 12 days after BCP. (**A–E**) Immunostaining images exhibit GPR160 protein expression (green) within a subset of DiI-labeled tibia sensory neurons (red) in rats ($n = 5$), Data are mean ± SEM of biological replicates. This expression coincided with the pan-neuronal marker NeuN (purple). Quantitative analysis was performed to determine the percentage of overlap. Scale bars: 100 μm. The white arrow highlights the cells NeuN labeled for GPR160 stained with DiI. (**F–T**) Triple immunostaining revealed the co-localization of GPR160 protein (green) in DiI-labeled neurons (red) with IB4 ($n = 5$ rats), CGRP ($n = 5$ rats), and NF200 ($n = 5$ rats), Data are mean ± SEM of biological replicates. Quantitative analysis was conducted to determine the percentage of overlap. Scale bars: 100 μm. The white arrow highlights the cells IB4, CGRP and NF200 labeled for GPR160 stained with DiI. Source data are available online for this figure.

anxiety-like behavior during evoked responses, indicated by the open-field test (specifically, a reduced duration of time spent and diminished distance traveled within the central area), as compared to AAV2/9-*Gfp* treated rats (Fig. EV4M–O). Viral injection exerted no discernible influence on locomotor functions (Appendix Table S1). Collectively, our observations suggest that mirroring the BCP-induced increase of GPR160 within the DRG elicits both anxiety-like behavior and enhanced nociceptive sensitivities.

## Global or conditional knockout of Gpr160 in nociceptive sensory neurons

In light of the potential off-target effects associated with siRNA or AAV-shRNA, we employed the CRISPR-Cas9 system to generate *Gpr160* mutant mice (designated as *Gpr160^{−/−}$) (Appendix Fig. S1A,B). Comparative analysis between the wild-type (WT) and *Gpr160^{−/−}$ mice revealed unaltered gross anatomy and immune organ morphology, including thymocytes, kidney, and spleen (Appendix Fig. S1C). The *Gpr160* ablation in DRG neurons was validated by

immunohistochemistry (Appendix Fig. S1D). Further investigation demonstrated unaltered cell size distribution patterns among CGRP, IB4, and NF200 markers in *Gpr160^{−/−}$ mice (Appendix Fig. S1E–J). These findings collectively affirm that the *Gpr160* mutation did not impart discernible effects on the DRG neuron distribution. Additionally, *Gpr160* gene knockout (KO) mice exhibited no impairment in motor function, baseline nociception, or light-touch sensation when compared to WT mice (Fig. EV5A–G). We initiated the BCP model in both WT and *Gpr160^{−/−}$ mice via femoral infiltration of Lewis lung carcinoma (*LLC*) tumors (Fig. 4A,B). The hematoxylin–eosin staining (Appendix Fig. S2A), CT with 3-dimensional reconstruction (Appendix Fig. S2B), and open-field test (Fig. EV5H–J) were used to validate the BCP model. Notably, BCP elicited strong mechanical allodynia and heat hyperalgesia in WT mice but had no implications in *Gpr160^{−/−}$ mice (Fig. 4C–E). Locomotor functions remained unaffected in both WT and *Gpr160^{−/−}$ mice after BCP (Appendix Table S1). These findings emphasize the critical involvement of GPR160 in the development of pain-like behavior induced by BCP in mice without observable impact on baseline nociceptive responses.

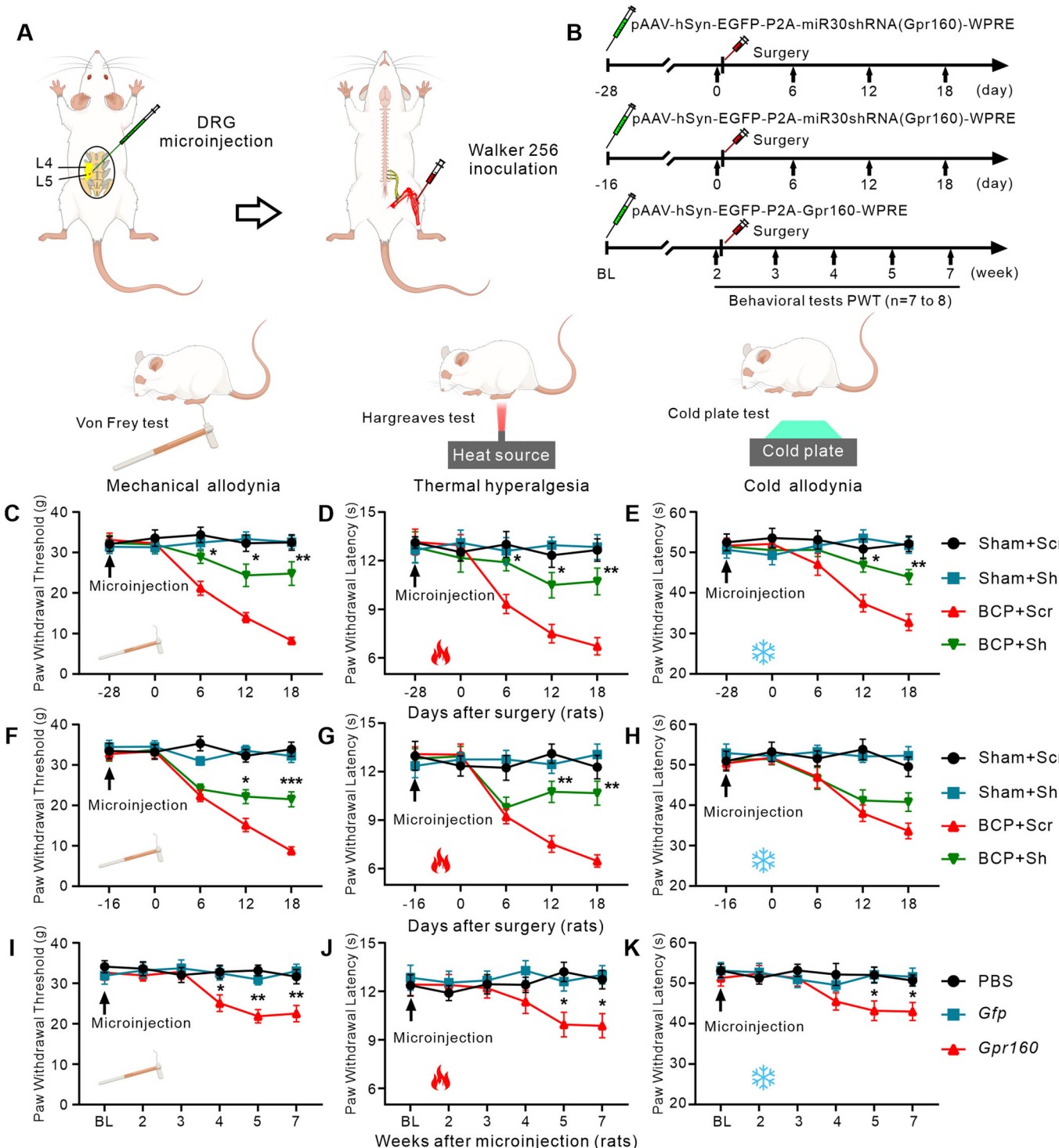

To further exploration of GPR160 function in peripheral sensory neurons, we created the *Gpr160*^fl/fl mice employing Cre-Loxp recombination system (Appendix Fig. S2C). Subsequently, *Gpr160*^fl/fl mice were generated to evaluate the impact of DRG *Gpr160* conditional knockout (cKO) on pain thresholds following DRG microinjection of rAAV-Cre with the *Pirt* promoter, a gene specifically expressed in peripheral sensory neurons. rAAV-*Gfp* served as the control. As anticipated, during the 4th-week post rAAV-*Pirt*-Cre injection into the DRG, the levels of DRG *Gpr160* mRNA and protein were reduced (Appendix Fig. S2D–F),

but not the spinal cord (Appendix Fig. S2G,H), as compared with the rAAV-*Pirt*-Gfp treated mice, indicating the efficiency and specificity of conditional knockout of *Gpr160* gene. Like the *Gpr160* siRNA, AAV-shRNA-treated rats and gko mice, *Gpr160*^fl/fl mice injected with rAAV-*Pirt*-Cre, but not rAAV-Gfp, exhibited both attenuated mechanical, thermal and cold hyperalgesia (Fig. 4F–H) on POD 6 to 18. Additionally, no changes in locomotor were observed in virus-injected *Gpr160*^fl/fl mice (Appendix Table S1). Collectively, our data strongly support the pivotal role of DRG GPR160 in BCP pathogenesis.

◄ **Figure 3. DRG increased GPR160 is required for development and maintenance of bone cancer pain in rats.**

(A, B) Experimental time line of rats BCP, AAV microinjection, and pain behavioral testing. (C–E) The impact of pre-microinjection with AAV-*Gpr160*-shRNA (shRNA) or control AAV-scramble-shRNA (Scr) into the ipsilateral L4/5 DRG on the development of BCP-induced mechanical allodynia (C), heat hyperalgesia (D), and cold allodynia (E) was evaluated 28 days prior to surgery on the ipsilateral side of rats. Data are mean ± SEM of biological replicates $n = 7$–8 rats/group. mechanical allodynia (*$P = 0.0245$, *$P = 0.0346$, **$P = 0.0041$), heat hyperalgesia (*$P = 0.0263$, *$P = 0.0416$, **$P = 0.0094$), cold allodynia (*$P = 0.0208$, **$P = 0.0066$) versus the BCP plus Scr group at the corresponding time points by two-way ANOVA with repeated measures followed by post hoc Tukey test. (F–H) Microinjection of the DRG with AAV-*Gpr160*-shRNA (shRNA) or control AAV-scramble-shRNA (Scr) 16 days before surgery alleviated the persistence of BCP-induced mechanical allodynia (F), heat hyperalgesia (G), and cold allodynia (H). Data are mean ± SEM of biological replicates $n = 7$–8 rats/group. mechanical allodynia (*$P = 0.0459$, ***$P = 0.0004$), heat hyperalgesia (**$P = 0.0084$, **$P = 0.0021$) versus the BCP plus Scr group at the corresponding time points by two-way ANOVA with repeated measures followed by post hoc Tukey test. (I–K) Paw withdrawal responses on the ipsilateral side to mechanical (I), heat (J), and cold (K) stimuli were assessed at designated time points following the microinjection of AAV-*Gpr160* (*Gpr160*), AAV-*Gfp* (*Gfp*), or PBS into unilateral L4/5 DRGs in naive rats. Data are mean ± SEM of biological replicates $n = 7$–8 rats/group. mechanical allodynia (*$P = 0.0391$, **$P = 0.0022$, **$P = 0.0041$), heat hyperalgesia (*$P = 0.0398$, *$P = 0.0140$), cold allodynia (*$P = 0.0305$, *$P = 0.0378$), versus the *Gfp* group at the corresponding time points by two-way ANOVA with repeated measures followed by post hoc Tukey test. Source data are available online for this figure.

## GPR160 is required for DRG neuronal hyperexcitability induced by bone cancer pain

Recognizing the significant involvement of nociceptive DRG neurons in BCP, we employed patch-clamp recordings to assess the excitability of these neurons in cancer-bearing mice. Our observations revealed that nociceptive DRG neurons with a small diameter (<25 μm) (Fig. 5A), specifically the *Gpr160*−/− mice, exhibited notably elevated action potential (AP) threshold and rheobase (pA) compared to the control WT neurons sourced from mice with BCP (Fig. 5C,D). Surprisingly, the frequency and Number of Aps in *Gpr160*−/− mice were decreased relative to controls, respectively (Fig. 5F,G). Furthermore, no statistically significant variations were found in resting membrane potential (RMP) and AP amplitude (Fig. 5B,E). These findings suggest the indispensable role of GPR160 in tumor-induced hyperexcitability of nociceptive DRG neurons.

Subsequently, we sought to ascertain the sufficiency of GPR160 in modulating DRG neuron excitability. LV-*Gpr160*-GFP and control were transduced into cultured DRG neurons. We assessed the excitability of small-diameter nociceptive neurons (<25 μm) expressing GFP using whole-cell patch-clamp recordings (Fig. 5H). Upon exposure to various current stimuli, *Gpr160*-overexpressing DRG neurons exhibited a notable reduction in both the AP threshold and rheobase (pA) (Fig. 5J,K). Moreover, no statistically significant differences were observed in RMP and AP amplitude compared to control neurons within the DRG neurons exhibiting GPR160 overexpression (Fig. 5I,L). Notably, the frequency and count of APs substantially increased in *Gpr160*-overexpressing DRG neurons (Fig. 5M,N). However, there is a change in the rheobase between cells from Wt/KO-BCP mice and AAV-injected mice, which may be due to many other factors. Different batches of primary cultured neurons, different environmental temperatures for cell detection, etc. may affect the excitability of the neurons, thereby altering their response to electrical stimulation and leading to changes in rheobase (Viatchenko-Karpinski and Gu, 2018). Overall, our findings affirm that GPR160 within DRG neurons is pivotal for enhancing DRG neuronal excitability.

## Histone modifications induce a GPR160 increase in the rat model of bone cancer pain

How does GPR160 exhibit upregulation within the DRG subsequent to tumor infiltration? Epigenetic processes, encompassing DNA methylation, noncoding RNA modulation, and histone modifications, are

essential in governing gene transcription and expression implicated in the mechanisms underlying pain development and maintenance (Du et al, 2022; Jiang et al, 2022a; Pan et al, 2021; Qi et al, 2022). In order to elucidate the underlying mechanisms governing enhanced GPR160 expression in a rat model of BCP, we conducted an inquiry into potential alterations in DNA methylation, noncoding RNA profiles, and histone modifications. Upon discovery of a CpG island housing 10 CpG sites near the transcriptional initiation site within the *Gpr160* promoter region (Appendix Fig. S3A), bisulfite sequencing PCR (BSP) was employed to examine alterations in methylation within this locus in both Sham and BCP rat models. Regrettably, our comprehensive analyses did not reveal a statistically significant difference in methylation status within the CpG island between the aforementioned groups, thereby dismissing the likelihood of DNA methylation involvement in this regulatory process (Appendix Fig. S3B,C). Furthermore, employing the microRNA prediction tool TargetScan (https://www.targetscan.org), we identified GPR160 as a potential target gene for miRNA-199-3p (Appendix Fig. S3D). Notably, diminished expression of miRNA-199-3p has been correlated with both visceral pain and neuropathic pain (Feng et al, 2020; Zhou et al, 2016). Subsequently, we investigated the potential regulatory interaction between miRNA-199-3p and GPR160. However, our assessment of luciferase activity in PC12 cells transfected with the wild-type 3' untranslated region (3'UTR) of GPR160 did not reveal any discernible impact induced by miRNA-199-3p (Appendix Fig. S3E).

Recent findings underscore the critical role of histone modifications (Jiang et al, 2022b), particularly lysine acetylation and methylation, in the regulation of gene expression. Acetylation of lysine 27 on histone H3 (H3K27) facilitates gene transcription, while trimethylation of H3K27 is well recognized to be strongly associated with gene repression (Lee et al, 2020; Zhang et al, 2021b). Consequently, we quantified the levels of H3K27 trimethylation (H3K27me3) and H3K27 acetylation (H3K27ac) in cytoplasmic and nuclear proteins subsequent to tumor infiltration. Our findings demonstrated a significant reduction in H3K27me3 (Fig. 6A,B) and a concomitant increase in H3K27ac (Fig. 6C,D) within the nuclear protein fraction of ipsilateral DRG on POD12 compared to sham-operated rats. Subsequent immunostaining analysis of DRG sections revealed a co-localization of H3K27me3 and H3K27ac modifications with GPR160 in both sham-operated and BCP rats (Fig. 6E). Remarkably, the majority of DRG neurons in BCP rats exhibited heightened GPR160 expression when displaying decreased H3K27me3 levels. Conversely, those neurons displaying augmented levels of H3K27ac expressed heightened GPR160 in the BCP DRG (Fig. 6F). Furthermore, in

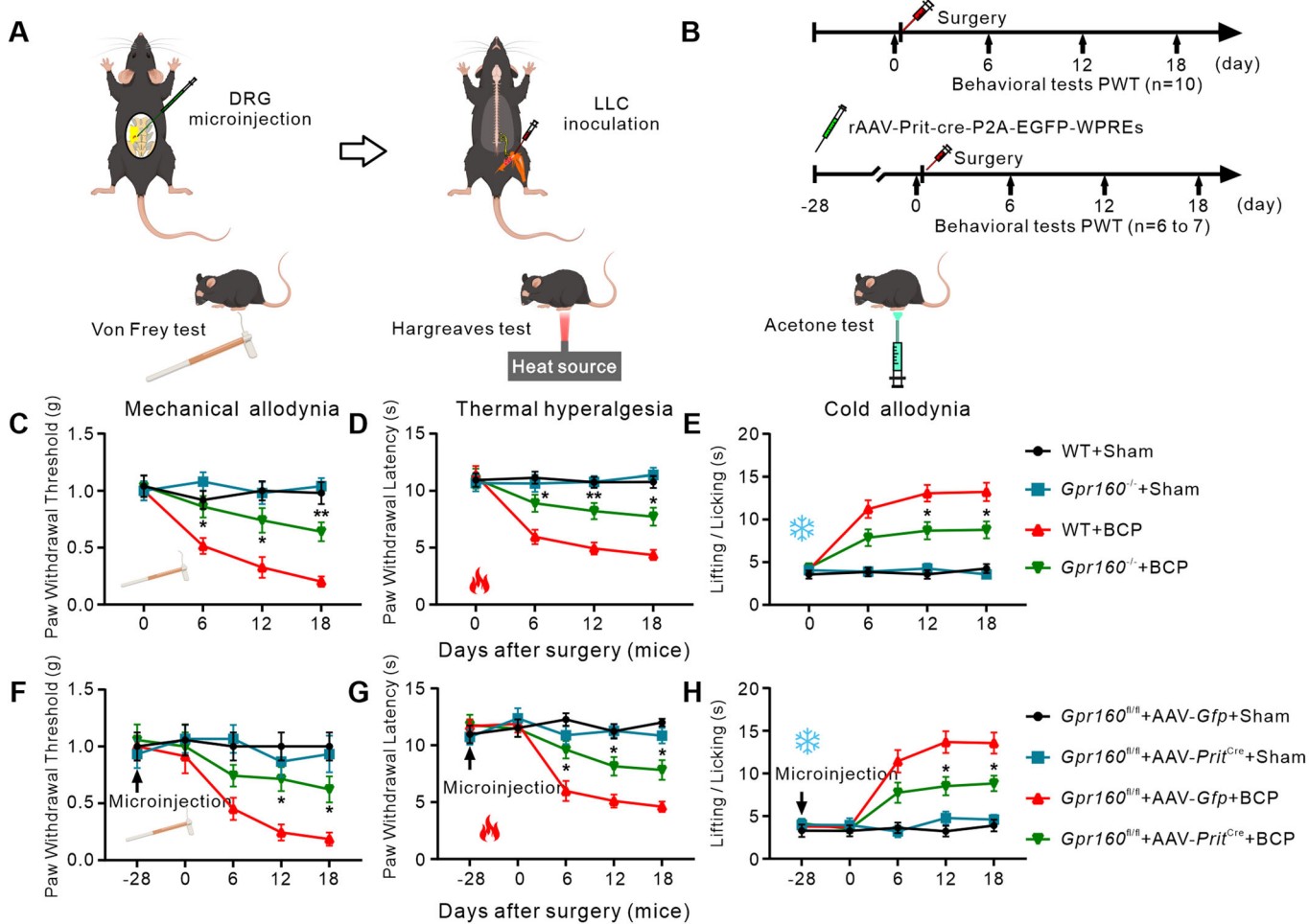

**Figure 4. Knockout of *Gpr160* in nociceptors alleviates bone cancer pain in mice.**

(A, B) Experimental time line of mice BCP, AAV microinjection, and pain behavioral testing. (C–E) Behavioral tests showing mechanical allodynia (C), heat hyperalgesia (D) and cold allodynia (E) in WT and *Gpr160*−/− mice. Data are mean ± SEM of biological replicates n = 10 mice/group. mechanical allodynia (*P = 0.0434, *P = 0.0413, **P = 0.0020), heat hyperalgesia (*P = 0.0353, **P = 0.0085, *P = 0.0105), cold allodynia (*P = 0.0280, *P = 0.0327), versus the WT plus BCP group at the corresponding time points by two-way ANOVA with repeated measures followed by post hoc Tukey test. (F–H) Effect of pre-microinjection of AAV-*Pirt*-cre or AAV-*Gfp* into the ipsilateral L3/4 DRG of *Gpr160*fl/fl mice on the development of BCP-induced mechanical allodynia (F), heat hyperalgesia (G), and cold allodynia (H) on the ipsilateral side. Data are mean ± SEM of biological replicates n = 6–7 mice/group. (*P = 0.0169, *P = 0.0330), heat hyperalgesia (*P = 0.0401, *P = 0.0457, *P = 0.0374), cold allodynia (*P = 0.0408, *P = 0.0441), versus the *Gpr160*fl/fl plus AAV-*Gfp* plus BCP group at the corresponding time points by two-way ANOVA with repeated measures followed by post hoc Tukey test. Source data are available online for this figure.

both sham-operated and BCP groups, H3K27me3 and H3K27ac were principally localized within the cell nuclei. In rat DRG lysates, ChIP analysis demonstrated amplifiable fragments within the *Gpr160* gene promoter from the immunoprecipitation complex with anti-H3K27me3 or anti-H3K27ac antibodies in sham-operated rats (Fig. 6G), affirming the targeted interaction of H3K27me3 and H3K27ac with the *Gpr160* gene. Notably, tumor infiltration contributed to a significant hindrance in the binding affinity of H3K27me3 and a concomitant increase in the binding affinity of H3K27ac, as reflected by band density, within the ipsilateral DRG on POD 12 in comparison to the levels observed in sham-operated rats (Fig. 6F,G). Subsequently, we investigated the potential role of diminished H3K27me3 and heightened H3K27ac levels in the mechanical allodynia observed on POD 12. Micro-administration of GSK-J4, a specific H3K27me3 agonist, markedly

intensified the reduction of H3K27me3 induced by BCP (Fig. EV5K,L). Correspondingly, administration of CBP/p300, a highly specific inhibitor of H3K27ac, led to a significant reduction in H3K27ac levels (Fig. EV5M,N), affirming its influence on the observed H3K27ac alterations. Subsequently, to elucidate the modulatory influence of H3K27me3 and H3K27ac on GPR160, we evaluated GPR160 expression after tumor infiltration. Remarkably, administration of GSK-J4 or CBP/p300 resulted in a significant reduction in GPR160 levels on POD 12. Remarkably, administration of both GSK-J4 and CBP/p300 produced a significant effect on the level of GPR160 compared to that of GSK-J4 or CBP/p300 alone (Fig. 6H,I). Furthermore, we examined the roles of H3K27me3 and H3K27ac in BCP in rats. Microinjec-tion of GSK-J4 partially attenuated mechanical allodynia on POD 12. Similarly, a single administration of CBP/p300 significantly

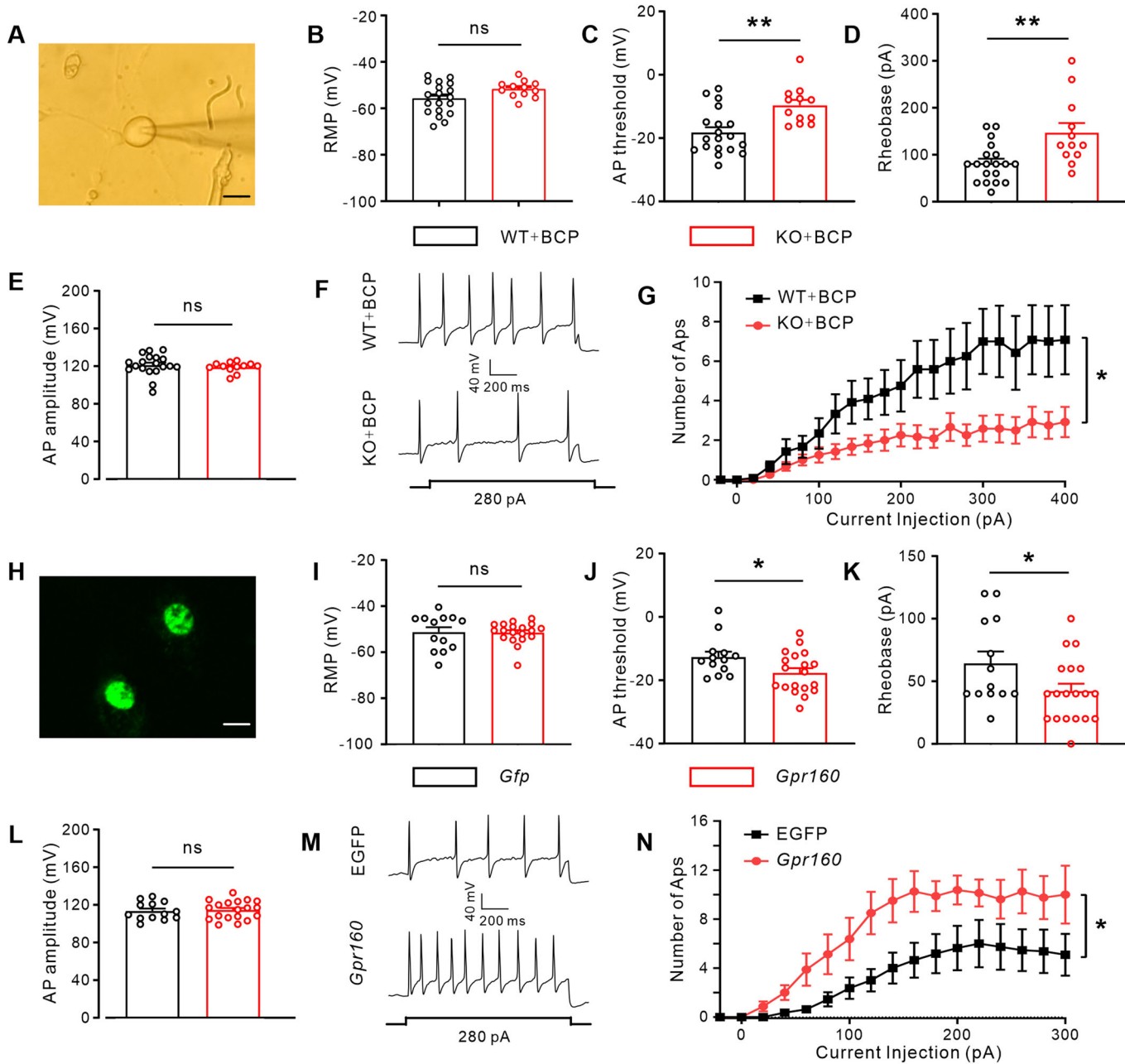

**Figure 5. GPR160 is critical for DRG neuronal hyperexcitability induced by tumor infiltration.**

(A) Image displaying an isolated DRG neuron with a pipette tip in the context of patch clamp recording. Scale bar: 20 μm. (B–E) Resting membrane potential (RMP) (B, P = 0.0612 versus WT plus BCP group), action potential (AP) threshold (C, **P = 0.0017 versus WT plus BCP group), current threshold (rheobase) (D, **P = 0.0034 versus WT plus BCP group), and after-hyperpolarization (AHP) amplitude (E, P = 0.5421 versus WT plus BCP group) were measured in DRG neurons from *Gpr160* KO mice and WT mice 12 days after BCP. Student's unpaired *t* test, Data are mean ± SEM of biological replicates n = 12–19 neurons from 3 to 4 mice. (F) Examples of action potential (AP) traces from both WT and *Gpr160* KO DRG neurons are depicted. (G) The analysis demonstrates that *Gpr160* KO reduces the number of APs elicited by tumor infiltration. *P = 0.0384, two-way ANOVA. Data are mean ± SEM of biological replicates n = 12–19 neurons from 3 to 4 mice. (H) Image displaying an isolated DRG neuron following viral transductions. Scale bar: 20 μm. (I–L) Resting membrane potential (RMP) (I, P = 0.9402 versus *Gfp* group), AP threshold (J, *P = 0.0352 versus WT plus BCP group), current threshold (rheobase) (K, *P = 0.0442 versus WT plus BCP group), and after-hyperpolarization (AHP) amplitude (L, P = 0.7561 versus *Gfp* group) were recorded from DRG neurons of naive mice following AAV-*Gpr160* and AAV-*Gfp* transductions. Student's unpaired *t* test, Data are mean ± SEM of biological replicates n = 13–19. (M) Examples of AP traces from DRG neurons of naive mice post-AAV-*Gpr160* and AAV-*Gfp* transductions. (N) GPR160 overexpression in mice DRG neurons increased the frequency of APs. *P = 0.0218, two-way ANOVA. Data are mean ± SEM of biological replicates n = 8–19 neurons from 3 to 4 mice. Source data are available online for this figure.

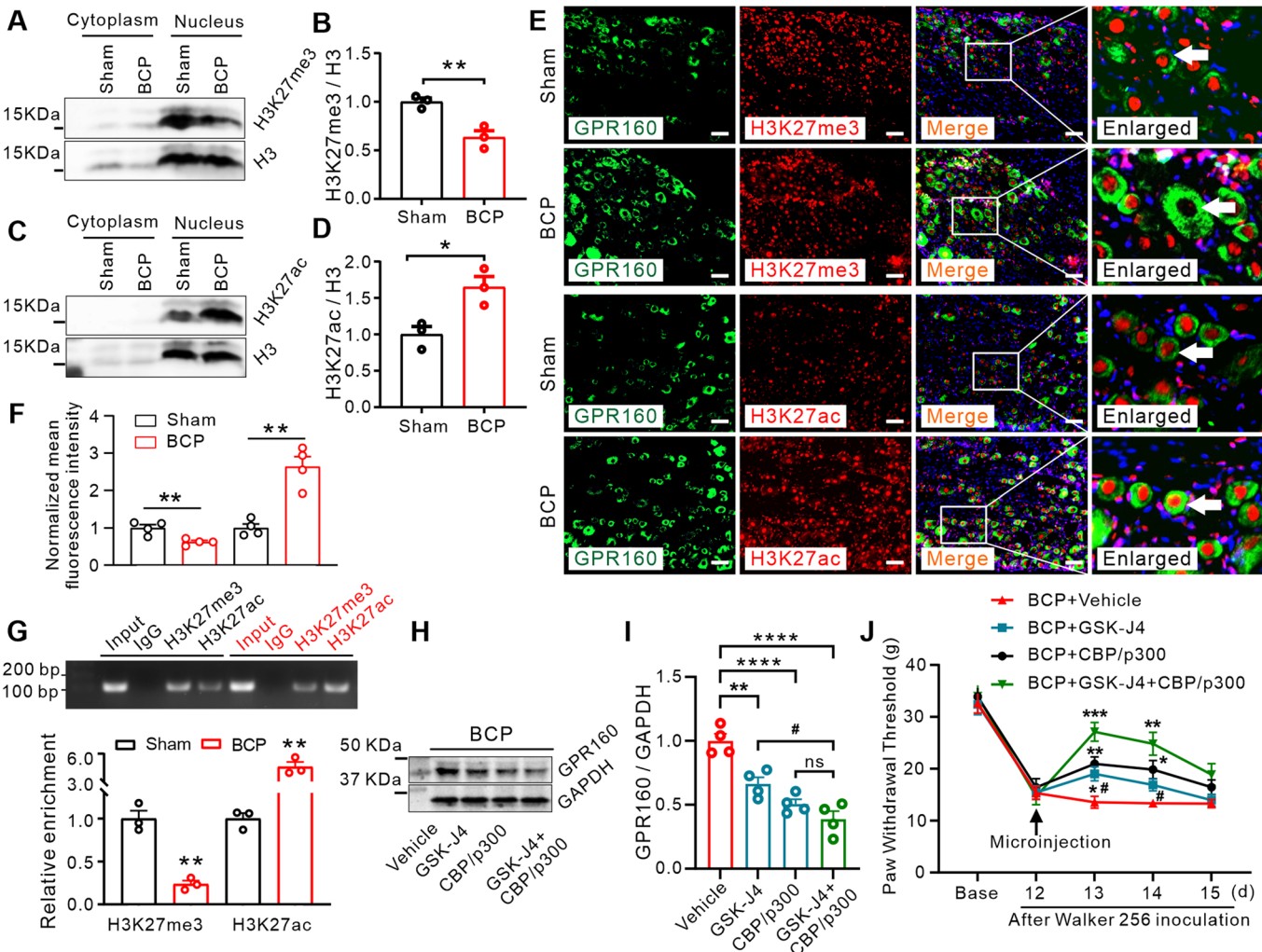

**Figure 6. Histone modifications induce GPR160 increased in the rat model of BCP.**

(A–D) Representative immunoblots (A, C) and corresponding summary data (B, D) showing the protein-expression levels of H3K27me3 and H3K27ac in DRGs on POD 12. Student's unpaired t test, Data are mean ± SEM of biological replicates n = 3 rats/group, **P = 0.0089, *P = 0.0228 versus sham group. (E, F) Double immunostaining of GPR160 (green) along with H3K27me3 or H3K27ac proteins (red) and DAPI (blue) was performed in rat DRGs from the BCP group compared to the sham-operated group on POD 12. Co-localization is indicated by white arrows (scale bar = 50 μm), with an enlarged view of the boxed area provided on the right. Student's unpaired t test, Data are mean ± SEM of biological replicates n = 4, **P = 0.0080, **P = 0.0013 versus sham group. The white arrow highlights GPR160⁺/H3K27me3⁺ and GPR160⁺/H3K27ac⁺ cells. (G) Representative blots and summarized ChIP-qPCR data demonstrate alterations in the binding of H3K27me3 or H3K27ac with Gpr160 following BCP. Input, total purified fragments. Student's unpaired t test, Data are mean ± SEM of biological replicates n = 3, **P = 0.0017, **P = 0.0024 versus sham group. (H, I) Representative immunoblots (H) and corresponding summary data (I) showing the impact of DRG microinjection of GSK-J4 and/or CBP/p300 on GPR160 expression in DRGs on POD 12. One-way ANOVA with repeated measures followed by post hoc Tukey test, Data are mean ± SEM of biological replicates n = 4 rats/group, **P = 0.0027, ****P < 0.0001 versus BCP plus vehicle group. #P = 0.0109, P = 0.3866 versus BCP plus GSK-J4 and CBP/p300 group. (J) Microinjection of GSK-J4 (5 nmol) and/or CBP/p300 (5 nmol) into the DRG mitigated mechanical allodynia in rats on POD 12. Data are mean ± SEM of biological replicates n = 7–8 rats/group. Two-way ANOVA with repeated measures followed by post hoc Tukey test. 13 d (*P = 0.0395, **P = 0.0059, ***P = 0.0004), 14 d (P = 0.0917, *P = 0.0357, **P = 0.0065) versus BCP plus vehicle. #P = 0.0190, #P = 0.0448 versus BCP plus GSK-J4 plus CBP/p300 group. Source data are available online for this figure.

elevated the mechanical pain threshold on POD 12 (Fig. 6J). Interestingly, on the 12th day following tumor infiltration, the combined microinjection of GSK-J4 and CBP/p300 demonstrated superior efficacy in suppressing mechanical allodynia in BCP rats compared to individual agent administration (Fig. 6J). Collectively, these observations indicate that tumor infiltration induces a reduction in H3K27me3 and an increase in H3K27ac modifications, consequently enhancing the transcriptional activation of GPR160 and contributing to the mediation of BCP.

## Histone modifications increase the binding of transcription factor Sp1 to GPR160 in bone cancer pain

Transcription factors are pivotal in governing gene transcription, and histone modifications can modulate this process by influencing transcription factor binding (Sinha et al, 2023). In our endeavor to ascertain the potential transcription factors of GPR160, we employed two computational algorithms, JASPAR (https://jaspar.genereg.net) and PROMO (http://alggen.lsi.upc.es), to

predict transcription factor binding sites within the *Gpr160* promoter region. The sequence from −2000 to 0 bp of the *Gpr160* promoter was analyzed. Our assessment revealed potential binding sites for two transcription factors, Specificity protein 1 (Sp1) and cAMP-responsive element binding protein 1 (CREB1), surpassing the defined 80% profile score thresholds (Fig. 7A). To experimentally validate the predicted interaction between the *Gpr160* promoter region and these transcription factors, luciferase reporter plasmids containing promoter sequences with predicted binding sites were constructed. Luciferase reporter assays demonstrated a significant increase in luciferase activity upon overexpression of Sp1 but not CREB1 (Fig. 7B). Further investigation into the crucial binding site region of Sp1 within the *Gpr160* promoter for gene expression involved a promoter truncation experiment, wherein the promoter was truncated in intervals of 500 bp based on predicted binding sites. The luciferase reporter assay results indicated that co-transduction of the full-length Sp1 vector with the GPR160 reporter vector notably enhanced Sp1 binding in the GPR160 promoter core region (−500–0 bp) (Fig. 7C). ChIP-qPCR analysis demonstrated specific amplification of a *Gpr160* gene promoter fragment within the immunoprecipitated complex using an anti-Sp1 antibody, while no amplification was observed from the complex immunoprecipitated with control IgG (Fig. 7D). This finding substantiates the precise binding of Sp1 to the *Gpr160* gene. Moreover, Sp1 binding to the *Gpr160* promoter region significantly intensified on POD 12 (Fig. 7D).

Sp1, a widely recognized transcriptional activator in regulating endogenous gene transcription linked to inflammatory effects (Park et al, 2016; Wei et al, 2022), was examined for its potential involvement in BCP. We assessed the expression of *Sp1* mRNA and protein to investigate Sp1's role in this context. Real-time PCR and western blot analyses manifested a significant elevation of Sp1 expression in response to tumor infiltration treatment on POD 6, 12, and 18 within DRG (Fig. 7E,F). Next, immunofluorescence analysis of GPR160 combined with Sp1 revealed endogenous Sp1 expression in sham rat DRG neurons (Fig. 7G). This analysis revealed a notable co-localization of Sp1 with GPR160 and an augmented expression of Sp1 in the BCP model (Fig. 7H). Immunostaining demonstrated that Sp1 was detected in both cytoplasm and nuclei, with a predominant nuclear localization as indicated by DAPI staining. Subsequent validation of *Gpr160* mRNA distribution within the DRG was conducted via RNAscope in situ hybridization (ISH). As shown in Fig. 7I, DRG *Rbfox3* mRNA positive for *Gpr160* mRNA co-localized with *Sp1* mRNA neurons. This prompted us to investigate whether the heightened GPR160 expression stemmed from altered Sp1 expression levels. To elucidate SP1's in vivo role, we administered *Sp1* siRNA via microinjection into the unilateral DRG on POD 12. Sp1 siRNA, but not scramble siRNA, effectively attenuated mechanical allodynia (Fig. 7K). RT-PCR validation confirmed a reduction in *Sp1* mRNA levels in the DRGs following knockdown on POD 14 (Appendix Fig. S3F). Western blot analysis demonstrated a significant reduction in increased GPR160 expression following *Sp1* siRNA administration on POD 14 (Fig. 7J). Furthermore, a single dose of *Sp1* siRNA notably increased the mechanical pain threshold on POD 13 to 14 (Fig. 7K). These findings collectively implicate histone modification in enhancing Sp1 binding to the *Gpr160* gene's promoter region, consequently upregulating GPR160 expression.

## Discussion

In this study, we provide preliminary evidence that tumor infiltration leads to an upregulation of GPR160 expression through Sp1-mediated activation in injured DRG neurons. This upregulation is associated with a decrease in H3K27me3 expression and an increase in H3K27ac levels in the DRG, which culminates in the manifestation of BCP symptoms. Utilizing both rat and transgenic mouse models, in conjunction with detailed behavioral assessments and electrophysiological analyses, our research robustly demonstrates that GPR160, localized in nociceptors, plays a pivotal role in modulating BCP. These findings reveal a novel mechanism whereby enhanced Sp1 activity drives *Gpr160* gene transcription via histone modifications in nociceptive DRG neurons under conditions reflective of BCP in rodent models (Fig. 8).

Recent studies have highlighted the critical role of orphan G protein-coupled receptors (oGPCRs) within DRG neurons in the development and maintenance of pathological pain (Nicol et al, 2015; Xia et al, 2021). However, the specific function of oGPCRs in BCP remains unclear. Among the oGPCRs identified, GPR160 has emerged as one of the top eight upregulated genes in DRGs at 12 days post-tumor infiltration. Previous studies have reported that GPR160 is exclusively expressed in the rodent dorsal horn of the spinal cord following traumatic nerve injury (Yosten et al, 2020). Nevertheless, the functional significance and distribution pattern of GPR160 expressed in DRG neurons in the context of BCP remain unexplored.

Pain induced by noxious stimuli involves the activation of cutaneous Aδ and C nociceptors, which are the peripheral terminals of small-diameter nociceptive DRG neurons (Colloca et al, 2017; Peirs and Seal, 2016). Extensive research has documented the involvement of small-diameter C and Aδ fiber neurons in various pathological pain models (Jayaraj et al, 2018; Tsantoulas et al, 2017; Wang et al, 2021b). In our cohort of DiI-labeled tibia-innervating sensory neurons, GPR160 expression was detected in both small- and large-diameter neurons. Notably, there was a greater degree of co-localization with isolectin B4 (IB4) and calcitonin gene-related peptide (CGRP) compared to neurofilament 200 (NF200). Given the critical role of IB4 and CGRP neurons, which extend C-fibers in the peripheral nervous system and transmit noxious signals to the spinal cord (Jiang et al, 2021), it is important to consider the potential contribution of GPR160-expressing neurons in BCP. Genetic ablation of unmyelinated sensory neurons expressing the GPCR has been shown to significantly reduce both acute and persistent mechanical pain (Cavanaugh et al, 2009). However, the involvement of A-fiber GPR160 neurons in BCP cannot be dismissed, considering the well-established role of large Aβ nerve fiber firing in the development of mechanical allodynia (Zhen-Zhong Xu et al, 2015).

Our findings consistently demonstrate an upregulation of *Gpr160* mRNA and protein in injured DRG neurons following tumor infiltration, suggesting a significant role for GPR160 in the development and maintenance of BCP. Specifically, we show that microinjection of *Gpr160* siRNA or AAV-*Gpr160* shRNA into the DRG effectively attenuates BCP-induced nociceptive hypersensitivities. Importantly, the administration of AAV-*Gpr160* shRNA does not affect basal pain perception or locomotor function, corroborating previous reports that GPR160 inhibition does not alter baseline nociceptive responses (Yosten et al, 2020). Interestingly, microinjection of AAV-*Gpr160* shRNA into the DRG did not result in a

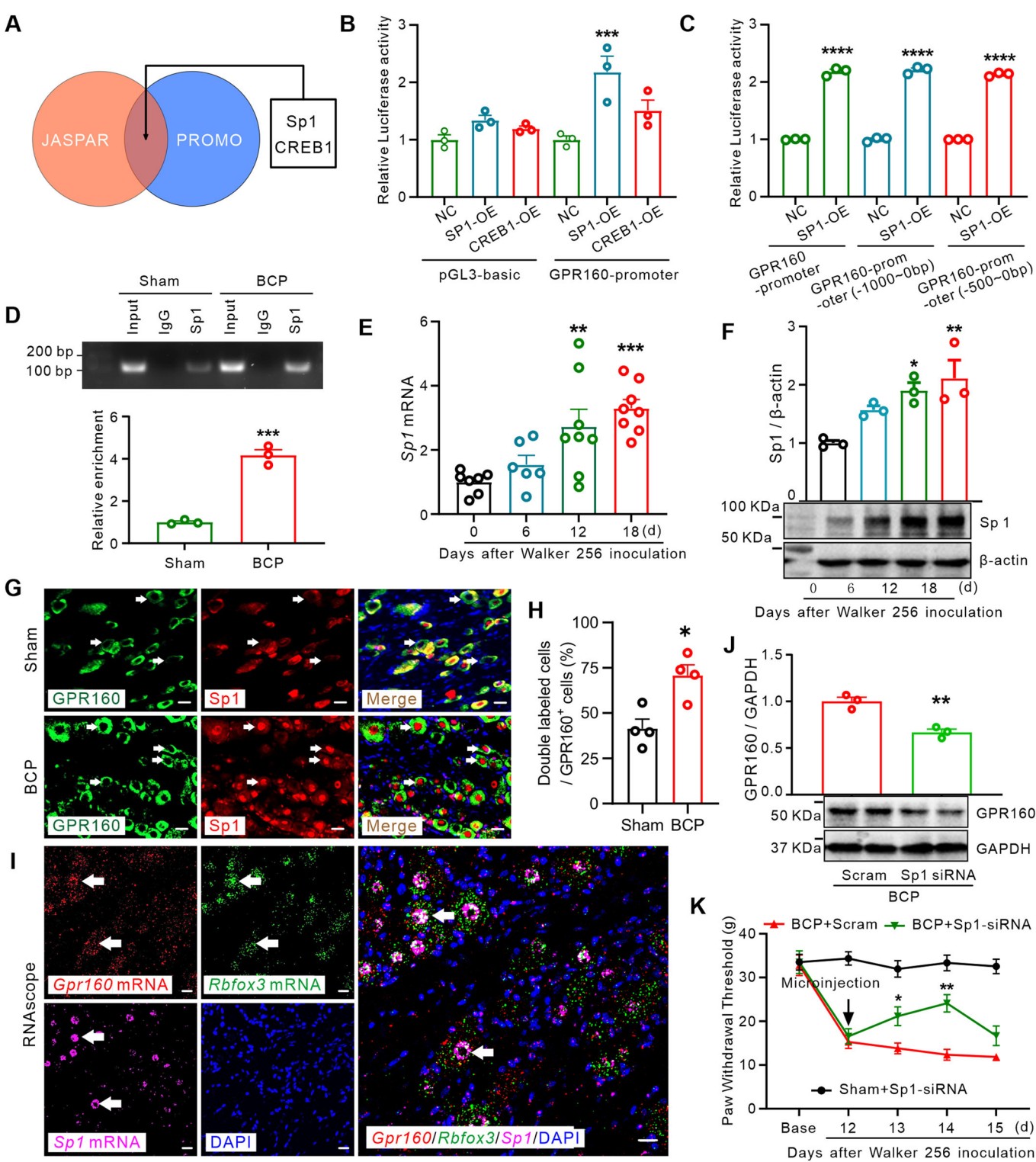

substantial reduction of basal GPR160 levels in the sham DRG, likely due to the relatively low baseline expression of GPR160 under normal conditions, which may not be sufficiently suppressed by the current dosage of AAV-*Gpr160* shRNA. Conversely, overexpression of GPR160 via DRG microinjection of AAV-*Gpr160* or LV-*Gpr160* resulted in enhanced sensitivity to mechanical and thermal stimuli,

as well as cold allodynia. Furthermore, our investigation revealed that the global knockout of *Gpr160* mitigated BCP without altering normal nociceptive thresholds, mirroring the observed phenotype in *Gpr160* cKO mice. Consequently, our findings underscore the pivotal involvement of GPR160 within tibia-specific nociceptive DRG neurons in the development and maintenance of BCP.

**Figure 7. Histone modification increases the binding of Transcription Factor Sp1 to *Gpr160* in BCP.**

(A) Venn diagrams demonstrate the prediction of binding of two transcription factors, Sp1 and CREB1, to the *Gpr160* promoter region using both the "JASPAR" (orange) and "PROMO" (blue) algorithms. (B, C) Dual-luciferase reporter assay was employed to validate the interaction between Sp1 and *Gpr160*. Subsequent experiments unequivocally identified Sp1 binding sites within the *Gpr160* gene, specifically located 500 bp upstream of the transcription start site ($-500$–$0$ bp). Student's unpaired $t$ test, Data are mean $\pm$ SEM of biological replicates $n = 3$, ***$P = 0.0005$, ****$P < 0.0001$ versus control group. (D) Representative ChIP-qPCR blots and summary data revealed an increased binding of Sp1 to the *Gpr160* gene promoter region following BCP. Student's unpaired $t$ test, Data are mean $\pm$ SEM of biological replicates $n = 3$, ***$P = 0.0003$ versus sham group. (E, F) The time course of Sp1 mRNA (E, **$P = 0.0059$, ***$P = 0.0004$ versus 0 d group) and protein (F, *$P = 0.0256$, **$P = 0.0077$ versus 0 d group) expression in the ipsilateral DRGs after BCP surgery. Data are mean $\pm$ SEM of biological replicates $n = 8/3$ rats/group. One-way ANOVA with repeated measures followed by post hoc Tukey test. (G, H) The double immunostaining of GPR160 (green), Sp1 (red), and DAPI (blue) in rat DRGs from the BCP group in comparison to the sham-operated group on POD 12. scale bar $= 20$ µm. Student's unpaired $t$ test, Data are mean $\pm$ SEM of biological replicates $n = 4$, *$P = 0.0103$ versus sham group. The white arrow highlights GPR160$^+$/Sp1$^+$ cells. (I) RNAscope ISH showing co-localization of DRG neurons (Rbfox3$^+$) expressing *Gpr160* mRNA with *Sp1* mRNA. Scale bar: 20 µm. The white arrow highlights Gpr160$^+$/Sp1$^+$/Rbfox3$^+$ cells. (J) Effects of Sp1 siRNA microinjection into DRGs on GPR160 protein expression on POD 15. Student's unpaired $t$ test, Data are mean $\pm$ SEM of biological replicates $n = 3$ rats/group, **$P = 0.0032$ versus BCP plus Scram group. (K) Sp1 siRNA DRG microinjection alleviated mechanical allodynia in rats on POD 12. Data are mean $\pm$ SEM of biological replicates $n = 7$–$8$ rats/group. Two-way ANOVA with repeated measures followed by post hoc Tukey test. *$P = 0.0349$, **$P = 0.0013$ versus BCP plus Scram group. Source data are available online for this figure.

The three primary epigenetic mechanisms regulating gene expression are DNA methylation, histone modification, and microRNA sponging(Jiang et al, 2022a; Liu et al, 2022; Qi et al, 2022; Wang et al, 2019). Our analysis found no alterations in DNA methylation at the CpG islands within the *Gpr160* promoter region, nor did we detect miRNAs directly targeting the 3'UTR of *Gpr160*. However, we identified significant changes in the occupancy of two key histone marks, H3K27me3 and H3K27ac, implicating their roles in the regulation of BCP in rats. H3K27me3 is well-established of transcriptional repression (Lee et al, 2020), whereas H3K27ac is associated with transcriptional activation (Zhang et al, 2021b). Previous studies have shown that both CFA injection and spinal nerve ligation (SNL) contribute to a reduction in H3K27me3 levels within the DRG, promoting the expression of TNF-α and IL-6, thereby contributing to the development and maintenance of inflammatory and neuropathic pain (Li et al, 2021; Qiao et al, 2023). Chronic postsurgical pain (CPSP) has also been shown to significantly increase H3K27ac levels while decreasing H3K27me3 at the promoter regions of key modulatory genes, such as Sprr1a and Anxa10, which are crucial in regulating neuropathic pain (Katsuda et al, 2021). Additionally, Qi et al reported increased H3K27me3 modifications following nerve injury, leading to the downregulation of miR-32-5p and the subsequent mediation of trigeminal neuropathic pain (Qi et al, 2022). In our investigation, we observed a significant reduction in H3K27me3 levels, accompanied by an increase in H3K27ac levels at the *Gpr160* promoter region in the DRG during BCP. Additionally, intervention with GSK-J4 or/and CBP/p300 significantly mitigated BCP-induced persistent hypersensitivity while concurrently reversing *Gpr160* expression. It is essential to consider that other mechanisms may also activate the Gpr160 gene in the DRG after BCP. For example, previous research showed that intrathecal administration of CART peptide (CARTp), acting as a GPR160 ligand, heightened pain sensitivity through GPR160-dependent ERK and CREB pathways (Yosten et al, 2020). However, Freitas-Lima et al later questioned GPR160's role as a receptor for CARTp (Freitas-Lima et al, 2023). Therefore, the precise mechanisms by which GPR160 contributes to pain require further investigation.

Histone methylation regulates transcription by modulating the binding of transcription factors (Qi et al, 2022; Torcal Garcia and Graf, 2021). Computational predictions utilizing Alibaba2 and PROMO algorithms identified potential binding sites for transcription factor CREB and Sp1, both linked to chronic pain (Lu et al,

2023; Yang et al, 2021) within the *Gpr160* gene's promoter region. Notably, only Sp1 overexpression significantly affected promoter luciferase activity, underscoring Sp1's essential role in *Gpr160* transcription. Sp1 has been shown to regulate calcium channels in neuropathic pain within the DRG (Gómez et al, 2019) and is a critical target in neuropathic pain pathogenesis (Miao et al, 2022). ChIP-qPCR analysis revealed increased Sp1 binding to the *Gpr160* promoter in DRG neurons under BCP conditions. This suggests that histone modifications, specifically reduced H3K27me3 and increased H3K27ac, may enhance Sp1 binding to the *Gpr160* promoter in the injured DRG.

In bone cancer, tumors and stromal cells within the bone marrow release chemical signals such as Nerve Growth Factor (NGF), Endothelin (ET), and hydrogen ions, which activate nociceptors, triggering electrical signals transmitted to the brain via the DRG and spinal cord (Yang et al, 2023; Zheng et al, 2022). Cancer-nerve crosstalk, another important mechanism, promotes tumor and neuronal growth, fostering a mutually beneficial interaction (Chu et al, 2022). BCP mechanisms involve both peripheral and central sensitization (Falk and Dickenson, 2014). Peripherally, BCP is driven by bone tissue injury and changes in the tumor microenvironment (TME) and DRG (Yang et al, 2024). Altered electrical excitability of DRG neurons, linked to voltage-gated and K$^+$ channels, plays a crucial role in this process. Calcium ions and TRP channels, key in nociceptive transmission, are modulated by GPCRs, influencing pain pathways (Xia et al, 2021; Yekkirala, 2013). GPR171, GPR65, GPR132, and GPR177 regulate TRP channels and are pivotal in pain (Xu et al, 2024). Downregulation of TRPV1 reduces Ca$^{2+}$ signals in spinal neurons, inhibiting wide dynamic range neuron activity and preventing BCP (Yin et al, 2024). ANO1 (TMEM16A), a Ca$^{2+}$-activated Cl$^-$ channel, is expressed in peripheral somatosensory neurons and interacts with TRPV1, selectively activated by GPCR-induced Ca$^{2+}$ release from intracellular stores (Jin et al, 2013). The role of GPR160 in BCP, potentially related to the TRPV1-ANO1 interaction, warrants further investigation.

Our study results indicate that GPR160 affects both the mechanical pain, cold allodynia and thermal hyperalgesia in BCP, which may be regulated by different mechanisms. Recent preclinical studies highlight the importance of mechanically sensitive signaling pathways, such as STING and PD-1/PD-L1 (Liu et al, 2020; Wang et al, 2021a), in the development and maintenance of BCP. These pathways, along with other

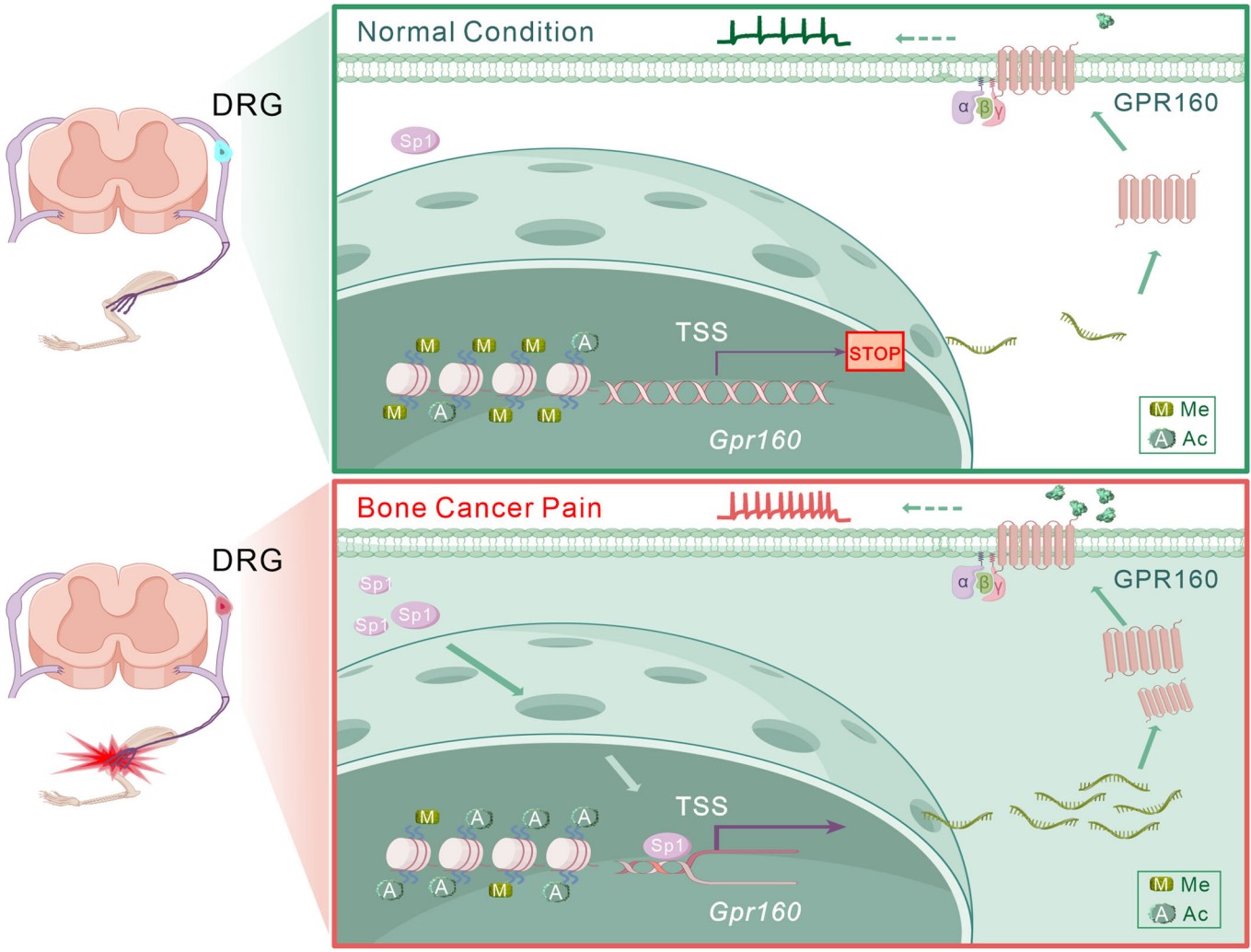

**Figure 8. Schematic diagram displaying the role and mechanism of GPR160 in bone cancer pain.**

The upper panel depicts a low GPR160 level in DRG neuron characterized by histone hypermethylation and hypoacetylation of the *Gpr160* promoter region under basal physiological conditions. In the lower panel, the intricate regulatory pathway of GPR160 in peripheral sensitization of neurons is elucidated. Initially, indeterminate intracellular signaling cascades modulate histone hypomethylation and hyperacetylation, facilitating the binding of the transcription factor SP1 to the *Gpr160* promoter, thereby inducing the heightened expression of GPR160. Subsequently, GPR160 instigates the reprogramming of neuronal gene expression, ultimately precipitating peripheral neuron sensitization and the onset of BCP. The schematic representation encapsulating these findings was generated utilizing Figdraw.

mechanisms like VEGF-A/VEGFR2-driven central sensitization and the TREK-1 potassium channel (Delanne-Cuménal et al, 2024; Hu et al, 2019), provide new insights and potential therapeutic targets for managing BCP. Cold and thermic-sensitive signaling mechanisms play an important role in pain. The transient receptor potential channels (TRP), also known as thermos TRP channels, intervene in the perception of hot and cold external stimuli. These channels, and especially TRPV1, TRPA1 and TRPM8, have been subjected to profound investigation because of their role as thermos sensors and also because of their implication in acute and chronic pain(Cabañero et al, 2022). Recent research has revealed that Sp1 functions as a cyclin-dependent kinase-5 (CDK5) activator and plays a pivotal role in inflammatory thermal hyperalgesia by impairing TRPV1 internalization (Adwan et al, 2015; Liu et al, 2019). It suggests that GPR160 may influence TRPV1

internalization. Therefore, it is essential to explore the mechanically, cold and thermic-sensitive signaling mechanisms of GPR160 in future studies.

In summary, our investigation uncovers a novel epigenetic mechanism by which GPR160 contributes to BCP in rodent models. We observed a significant and sustained upregulation of GPR160 in rat DRG neurons following tumor infiltration. This increase in GPR160 expression is associated with histone modification changes, notably a reduction in H3K27me3 and an increase in H3K27ac, as well as enhanced SP1 binding to the *Gpr160* promoter region. These molecular changes lead to the development of mechanical and thermal hyperalgesia. The identification of GPR160's regulatory role in BCP provides potential targets for the prevention and treatment of bone cancer-related pain.

# Methods

## Reagents and tools table

| Reagent/resource | Reference or source | Identifier or catalog number |
|---|---|---|
| **Experimental models** | | |
| SD rats (*R. norvegicus*) | Shanghai Laboratory Animal Centre (China) | N/A |
| C57BL/6 (*M. musculus*) | Shanghai Laboratory Animal Centre (China) | N/A |
| *Gpr160*⁻/⁻ (*M. musculus*) | GemPharmatech Co., Ltd | T006713 |
| *Gpr160*ᶠˡ/ᶠˡ (*M. musculus*) | GemPharmatech Co., Ltd | T006643 |
| Walker 256 cells (*R. norvegicus*) | Cell Culture Center of the Chinese Academy of Medical Sciences | N/A |
| LLC cells (*M. musculus*) | Chinese Academy of Sciences Cell Bank | SCSP-5252 |
| PC12 cell (*R. norvegicus*) | Chinese Academy of Sciences Cell Bank | TCR 8 |
| **Recombinant DNA** | | |
| pcDNA3.1(+)-Sp1-3Flag | This study | N/A |
| pcDNA3.1(+)-Creb1-3Flag | This study | N/A |
| pcDNA3.1(+)-Gpr160-3Flag | This study | N/A |
| **Antibodies** | | |
| Anti-GPR160 | USBiological | 223028 |
| Anti-GPR160 | Alomone labs | AGR-090 |
| Anti-H3K27me3 | Abcam | ab6002 |
| Anti-H3K27me3 | Abcam | ab192985 |
| Anti-H3K27ac | Abcam | ab4729 |
| Anti-H3K27ac | Thermo Scientific | MA5-42545 |
| Anti-Histone H3 | Abcam | ab1791 |
| Anti-RBFOX3/NeuN | NOVUS | NBP1-92693 |
| Anti-Glutamine Synthetase | Abcam | ab64613 |
| Anti-NF200 | EMD Millipore Corp | MAB5266 |
| Isolectin B4 | SIGMA | L2140 |
| CGRP | SIGMA | C7113 |
| Anti-SP1 | Proteintech | 21962-1-AP |
| Anti-SP1 | Proteintech | 66508-1-Ig |
| Anti-β-actin | HUABIO | R1207-1 |
| Anti-GAPDH | Affinity | AF7021 |
| Donkey Anti-Mouse (488) | Abcam | ab150105 |
| Donkey Anti-Mouse (594) | Abcam | ab150108 |
| Donkey Anti-Rabbit (405) | Abcam | ab175651 |

| Reagent/resource | Reference or source | Identifier or catalog number |
|---|---|---|
| Donkey Anti-Rabbit (488) | Abcam | ab150073 |
| HRP-Goat Anti-Mouse IgG | Biosharp | BL001A |
| HRP-Goat Anti-Rabbit IgG | Biosharp | BL003A |
| **Oligonucleotides and other sequence-based reagents** | | |
| Primers | This study | Appendix Table S2 |
| **Chemicals, enzymes and other reagents** | | |
| SYBR Premix Taq™ | TaKaRa | RR036A-1 |
| Fetal bovine serum | Gibco | 10091-148 |
| 0.5MEDTA | Beyotime Biotechnology | ST066 |
| 1.5MTris-HCL | Beyotime Biotechnology | ST789 |
| Trizol | Ambion | 15596026 |
| GPR160 Blocking Peptide | Alomone labs | AGR-090 |
| Lipofectamine 2000 | Invitrogen | 11668-019 |
| GSK-J4 | MCE | HY-15648B |
| CBP/p300 | MCE | HY-132197 |
| DiI | Sigma-Aldrich | D282 |
| Triton X-100 | BioFROXX | 1139ML100 |
| Phosphatase inhibitor | NCM Biotech | P002 |
| Opti-MEM® I | GIBCO | 51985-034 |
| Tween-20 | Solarbio | T8220 |
| EDTA | ThermoFisher | 17892 |
| Bovine serum albumin (BSA) | Sigma | A7030 |
| Dual-Luciferase Reporter Assay System | Promega | E1910 |
| PMSF | Sigma | 329-98-6 |
| **Software** | | |
| ImageJ | ImageJ Software | https://imagej.net/software/fiji/ |
| GraphPad Prism 9.0 | GraphPad Software | https://www.graphpad.com/ |
| **Other** | | |
| RNAscope assay | Advanced Cell Diagnostics | 323100 |
| AAV-hSyn-*Gpr160*-EGFP | Obio Technology | HY2111533 |
| AAV-hSyn-*Gpr160*-shRNA | Obio Technology | HY20234017 |
| AAV-Prit-cre-EGFP | BrainVTA | PT-7419 |
| SimpleChIP® Plus Enzymatic Chromatin IP Kit | Cell Signaling Technology | 9004S |
| PVDF membranes | Thermo Scientific | 88520 |

## Experimental animals

Male and female Sprague Dawley rats weighing between 180 and 200 g, as well as C57BL/6 mice aged 6 to 8 weeks, were acquired from the Shanghai Laboratory Animal Centre (China). *Gpr160$^{-/-}$* and *Gpr160$^{fl/fl}$* mice were engineered using the CRISPR/Cas9 system by GemPharmatech Co., Ltd in China. The animals were kept in a controlled setting where they were exposed to a 12-h cycle of light and darkness. They had unrestricted access to food and water. Also, they had environmental parameters maintained at 22–25 °C with humidity levels at 55–65%. Subsequently, the participants were randomized to the surgery and therapy groups in a random manner. Ethical clearance for all experimental procedures was obtained from the Institutional Animal Care and Use Committee of Jiaxing University in Jiaxing, China (NO. JUMC2023-038). Furthermore, the research adhered to the guidelines established by the International Association for the Study of Pain. To ensure impartiality in behavioral assessments, the researchers performing these studies were unaware of the specific model circumstances and pharmacological treatments. The trials were done throughout the time frame of 9:00 am to 6:00 pm.

## Reagents

The assays for BCA proteins were procured from ThermoFisher Scientific. The reverse transcription reagent and SYBR Premix Taq™ were obtained from TaKaRa, whereas Shanghai Sangon Co., Ltd. synthesized the primers. AAV -*Gpr160*-EGFP (titer $7.26 \times 10^{12}$ v.g./ml, catalog number: HY2111533), AAV-EGFP (titer $8.51 \times 10^{13}$ v.g./ml, catalog number: HY2111533), AAV-hSyn-*Gpr160*-shRNA (titer $1.70 \times 10^{13}$ v.g./ml, catalog number: HY20234017), and AAV-SYN-shRNA (titer $2.03 \times 10^{13}$ v.g./ml, catalog number: HY20234017) were produced by Obio Technology (Shanghai) Corp., Ltd. GSK-J4 and CBP/p300 were acquired from MCE and solubilized in 1% DMSO. Plasmids and siRNA were procured from Genepharma (Shanghai, China). AAV-Prit-cre-EGFP (titer $5.77 \times 10^{12}$ v.g./ml) was provided by BrainVTA (BrainVTA Co., Ltd., Wuhan, China).

## Cell culture

### Walker 256 cells
Walker 256 mammary carcinoma cells were obtained from the Cell Culture Center of the Chinese Academy of Medical Sciences (Beijing, China) and cultured in DMEM with 10% FBS at 37 °C in 5% $CO_2$. To induce ascites, 0.5 mL of Walker 256 cell suspension ($5 \times 10^6$ cells/mL) was injected intraperitoneally into rats (60–80 g). The ascitic fluid was collected, centrifuged, and washed three times with ice-cold PBS, with cells resuspended at $1 \times 10^5$ cells/μL.

### LLC cells
Murine Lewis lung carcinoma (LLC) cells, derived from a spontaneous carcinoma in C57BL/6 mice, were sourced from the Chinese Academy of Sciences Cell Bank, Type Culture Collection (Shanghai, China). These LLC cells were cultured in DMEM supplemented with 10% fetal bovine serum (Gibco, ThermoFisher Scientific), penicillin and streptomycin. The cultures were maintained at 37 °C in an atmosphere of 95% air and 5% $CO_2$.

## Bone cancer pain model

### Rat BCP model
To establish the rat BCP model, we followed a modified version of the procedure outlined in our prior investigation (Ni et al, 2023). Briefly, rats were initially anesthetized using 2–3% isoflurane and subsequently interstitially injected with 10 μl ($1 \times 10^6$ cells) using a microinjection syringe. In the sham control group, the rats were injected with 10 μl of heat-inactivated cells into their left tibia. Following a duration of roughly 1 min, the needle was retracted, and the injection hole was closed using bone wax. In order to retrogradely label tibial sensory afferents, DRG cell bodies along with their afferent fibers innervating tibia were facilitated through the utilization of a retrogradely transported red fluorescent dye, DiI (Sigma-Aldrich). DiI was injected into the bone marrow cavity of the tibia (2.5 mg/mL; 8 μL in 25% ethanol) at 12 days before harvesting.

### Mouse BCP model
LLC cells were enzymatically dissociated using 0.05% trypsin and prepared as a suspension at a concentration of $5 \times 10^7$/L in phosphate-buffered saline (PBS). The mouse model of bone cancer-induced pain (BCP) was established in accordance with the previously published protocol (Wang et al, 2020). Mice were anesthetized with 2–3% isoflurane and subsequently intrafemorally injected with 5 μl ($2.5 \times 10^6$ cells) using a 10 μl microinjection syringe. In the sham control group, 5 μl of heat-inactivated cells were injected into the left femoral cavity of the mice. Animals experiencing failed injections or displaying impaired mobility following tumor cell inoculation were excluded from the study.

## Behavioral tests

Animals were accustomed to the test environment for 30 min each time for 3 days prior to testing and were maintained at appropriate humidity and temperature each day. All the behavioral tests were conducted by individuals who were unaware of the therapy or genotypes of the animals. There was a 1–2-h delay between the two tests.

### Mechanical pain test
For rats, PWT was assessed using a previously described calibrated set of *Von Frey* monofilaments (BME-404, Institute of Biological Medicine, Academy of Medical Science, China) (Ni et al, 2023). The average of five consecutive measurements was taken as PWT.

For mice, Von Frey filaments (Stoelting, USA) were used to provide a series of stimuli to the middle area of the hind paw's plantar surface. The PWT was determined by measuring the weight in grams of the filament that caused withdrawal of the hind paw in three out of the five consecutive administrations (Zhang et al, 2022).

### Thermal tests
For rats and mice: The Hargreaves test used the Hargreaves radiant heat apparatus, specifically the IITC Life Science model, to conduct the radiant heat test. Each trial being repeated three times, with a 5-min gap between each repetition. To prevent any possible harm to the animal tissue, a time limit of 20 s was implemented. For the

hot plate test, thermal sensitivity was assessed utilizing a hot plate apparatus (ZS Dichuang, Beijing, China) with varying surface temperatures of 50, 52, or 56 °C. The latency to a nociceptive response, which encompassed behaviors such as licking, hind paw shaking, or jumping, was recorded. To mitigate the risk of tissue damage, distinct cutoff times were implemented: 60, 40, or 20 s for assays conducted at 50, 52, or 56 °C, respectively.

### Cold test

For rats: The cold plate test included assessing the paw withdrawal latencies of rats to noxious cold (0 °C). This was done by putting the rat in a specialized Plexiglas container that was positioned on a temperature-monitored aluminum plate that was chilly (ZS Dichuang, Beijing, China). The experiment included doing the test three times with a 10-min gap between each trial for the ipsilateral paw. To mitigate the risk of possible tissue injury in rats, a time limit of 60 s was imposed (Li et al, 2020).

For mice: Cold sensitivity was determined by employing the acetone test (Wang et al, 2020). The mice were positioned on an elevated metal mesh surface, and a syringe was employed to apply a droplet of acetone to the plantar hind paw. Subsequently, the duration during which the animals lifted or licked the paw within a 90-second timeframe was meticulously recorded.

### CatWalk gait analysis

For rats: This test was performed to assess pain-associated behaviors (Hu et al, 2019). To summarize, rats were placed at the open end of a glass platform that was enclosed. The platform was lit by a red ceiling light-emitting diode lamp. The rats were given the freedom to move around on the pathway. The high-speed camera positioned underneath the equipment collected photographs of the lighted portion of each paw as the rats moved over the glass floor. Valid data was defined as a minimum of four consecutive step cycles or full runs through the tunnel. The data were measured as the proportion of hind paw participation on the same side (ipsilateral) or opposite side (contralateral).

The open field test was used to assess the innate exploratory motor activity and anxiety-like behavior of both rats and mice (Ni et al, 2023; Zhang et al, 2023). A black behavior test box for rats (100 × 100 × 50 cm) and a white for mice (40 × 40 × 40 cm) was divided into central and peripheral fields and allowed to freely explore for a period of 10 min. Movements were recorded with a video camera placed above the apparatus and subsequently analyzed using Jiliang software (Shanghai, China). After each test session with a rat or mouse, fecal samples were collected, and the test area underwent thorough cleaning with a cloth containing 10% alcohol. The area was then dried with a cloth prior to the commencement of subsequent experiments.

Locomotor functions encompass placing, grasping, and righting reflexes in accordance with established procedures (Li et al, 2020). In brief, (i) Placing reflex: The hind limbs were situated somewhat below the forelimbs, and the upper surfaces of the hind paws touched the edge of a table. The act of placing the rear paws on the tabletop was seen and documented. (ii) Grasping reflex: upon placement of the animal on a wire grid, any reflexive grasping of the wire by the hind paws was recorded. (iii) Righting reflex: the animal's ability to promptly resume an upright position after being laid on its back on a flat surface was documented. The experiment was repeated five times with a time gap of 5 min between each trial.

The scores for each reflex were calculated by counting the number of normal responses seen.

For the tail-immersion assay, the tail-immersion assay was employed to evaluate the thermal sensitivity of mice. In this evaluation, one-third of the tail length was immersed in a water bath, which was consistently maintained at temperatures of 48 °C or 52 °C. We quantified the latency to a nociceptive response, specifically tail withdrawal. Each trial underwent three repetitions with 1-min intervals. To prevent tissue damage, cutoff times of 20 s and 10 s were implemented for the assays conducted at 48 °C and 52 °C, respectively.

For motor function test, a rotarod system (UgoBasile, Varese, Italy) was conducted to evaluate the motor function. The rotation began at 4 rpm and gradually accelerated to 40 rpm over a 5-min duration. Each trial was iterated three times with 10-min intervals between trials, and we recorded the latency at which the animal fell from the rod.

For flick hair skin test, the fur on the nape of the neck was carefully shaved prior to testing. On the testing day, mice were subjected to stimulation using a 0.07 g von Frey hair applied to the shaved nape skin for a total of 10 applications, with 1-min intervals between each trial. The resulting response behaviors, which included head shaking or scratching, were meticulously quantified.

For the pinprick test, mice were positioned on a heightened platform with a metal mesh floor and acclimated for 30 min. Subsequently, we gently applied an insect pin affixed to a 1 g von Frey filament to stimulate the plantar surface of the hind paw, ensuring no skin penetration occurred. The withdrawal response was quantified for a total of 10 trials, each separated by 1-min intervals between trials.

For cotton swab test, a cotton swab was expanded to three times its original size. Subsequently, we gently brushed the hind paw of the mice in a heel-to-toe manner, employing the enlarged cotton swab. This trial was replicated ten times, with 1-min intervals between each trial. Responses were quantified based on the occurrence of paw flicking or withdrawal in response to the stimulation.

## Bisulfite sequencing PCR

Genomic DNA was extracted and bisulfite-treated following the MSP method's described protocol. PCR amplification of the GPR160 promoter CpG island fragment from bisulfite-converted genomic DNA utilized primers detailed in Appendix Table S2. Subsequently, PCR products were purified using the QIAquick Gel Extraction Kit (Qiagen) and ligated into the pGEM-T Easy Vector (Promega) for sequencing. Ten colonies per mouse were randomly selected for subsequent sequencing.

## mRNA microarray

The expression profile of 22837 protein-coding transcripts was conducted with the Arraystar Rat LncRNA Microarray V2.0. Sample labeling and array hybridization were performed on the AgilentArray platform. In brief, total RNA from each sample was amplified and transcribed into fluorescent cRNA (Agilent Technologies, Palo Alto, USA) before 1 μg of each labeled cRNA was hybridized onto the microarray slide. The hybridized arrays were subsequently washed, fixed, and scanned using the Agilent Microarray Scanner. Array images so collected were studied with the Agilent Feature Extraction

software (version 10.5.1.1). We utilized the GeneSpring GX V11.5 software package (Agilent Technologies) to conduct quantile normalization and process the data. The differentially expressed novel oGPCR mRNAs with statistical significance were identified using heatmap and volcano plot filtering. The threshold used to screen upregulated mRNAs was a fold change ≥2 and $P < 0.05$. All raw fast files and the expression count matrix have been deposited into the National Center for Biotechnology Information's Gene Expression Omnibus (accession number GSE270839).

## DRG microinjection

DRG microinjections were conducted in accordance with the previous protocols (Luo et al, 2019). The lumbar articular process was exposed and partially excised with a cranial drill under microscopic observation. The surgical site was meticulously cleaned of surrounding tissue and blood using micro forceps and cotton swabs, thereby exposing the L4 and L5 DRG. Afterward, the DRG that was visible was injected with solutions (1–1.5 µl in rats and 0.5-1 µl in mice) using a glass micro-pipette linked to a Hamilton syringe. We selected Adeno-associated virus (AAV) with a titer of ≥$10^{13}$ vector genomes/ml, Lentivirus (LV) with a titer of ≥$10^8$ transduction units/ml, and siRNA at a concentration of 20 µM for our study, relying on established precedents (Ma et al, 2021). The TurboFect in vivo transfection reagent (ThermoFisher Scientific) was employed as a delivery vehicle to enhance delivery and prevent RNA and DNA degradation, as previously reported (Ma et al, 2021). The microsyringe pump was withdrawn after a 10-min injection period to facilitate virus diffusion. Afterward, the mice were given time to recuperate in a heated setting prior to being placed back in their original cage. Any animals exhibiting symptoms of paresis or other abnormalities were not included in the research. Hematoxylin/eosin staining was performed on the injected DRGs to evaluate structural integrity and confirm the absence of visible leukocytes.

## DRG neuronal culture and transduction

Mouse DRGs were meticulously isolated from 6 to 8-week-old mice and underwent viral transfection, adhering to established protocols (Li et al, 2020). Following euthanasia employing isoflurane, all DRGs were collected and placed in a cold Neurobasal Medium that was enriched with 10% fetal bovine serum (Gibco/ThermoFisher Scientific), 100 units/ml of penicillin, and 100 µg/ml of streptomycin. Afterward, the DRGs were treated with an enzyme solution containing 5 mg/ml of dispase and 1 mg/mL of collagenase type I in Hanks' balanced salt solution, which did not include $Ca^{2+}$ and $Mg^{2+}$. After grinding and centrifugation, the separated neurons were mixed with a combination of Neurobasal Medium and placed into six-well plates that had been treated with a layer of poly-D-lysine at a concentration of 50 µg/ml. The cells were cultured at a temperature of 37 °C in an environment consisting of 95% $O_2$ and 5% $CO_2$. On the 2nd day, a volume of 2–10 µl of virus, at a concentration of at least $1 \times 10^{12}$ GC/ml, was added to each well. The neurons were gathered 3 days later.

## Micro-CT

Tibias obtained from rats and femurs obtained from mice were preserved in a solution of 4% paraformaldehyde. They were then subjected to scanning using a micro-CT system (SkyScan 1276, Bruker, Belgium) (Chen et al, 2022). The specifications consisted of a current of 100 µA, a voltage set at 80 kVp, a pixel size of 20 µm, and an exposure period of 926 ms. Following the scanning process, 3D pictures of the distal metaphysis were produced.

## Bone histology

The rats/mice were anesthetized deeply and then underwent transcardial perfusion with a 4% solution of paraformaldehyde. Subsequently, tibia/femur bone was extracted and subjected to a 24-h decalcification process. After rinsing and dehydration, the bones were immersed in paraffin and then cut into 10 µm thick pieces employing a rotary microtome. Subsequently, these segments were stained using hematoxylin and eosin to facilitate the visualization of tumor infiltration and evaluate the extent of bone damage.

## Western blotting

Animals underwent anesthesia via isoflurane inhalation overdose, after which DRG tissues (L4/5 in rats and L3/4 in mice) were expeditiously dissected and homogenized in ice-cold lysis buffer. To ensure sufficient protein yield, tissue samples were combined from either two unilateral rat DRGs or four unilateral mouse DRGs. The supernatants were produced by centrifuging the mixture at 12,000 rpm at a temperature of 4 °C for a duration of 10 min. The protein content was measured using a BCA assay kit. Equivalent protein quantities were loaded, separated on SDS-PAGE gels, and then transferred to PVDF membranes. The membranes were obstructed using Protein-Free Rapid Blocking Buffer (Epizyme, China) for a duration of 30 min at ambient temperature. Primary antibodies included rabbit anti-GPR160 (1:1000, USBiological, 223028), mouse anti-H3K27me3 (1:1000, Abcam, ab6002), mouse anti-H3K27ac (1:1000, Abcam, ab4729), rabbit anti-Sp1 (1:1000, Proteintech, 21962-1-AP), rabbit anti-GAPDH (1:2000, Affinity, AF7021), and mouse anti-Histone H3 (1:1000, Abcam, ab1791). Signals were detected via enhanced chemiluminescence and captured using a ChemiDoc XRS system (Bio-Rad). The western blot analysis was conducted on three to four occasions, resulting in consistent results. The Bio-Rad image analysis equipment was employed to quantify the integrated optical density of certain bands, and blot intensities were quantified using ImageJ software.

## Reverse transcription-PCR

Quantitative real-time reverse transcription PCR (RT-PCR) was conducted according to established procedures. Total RNA was extracted from DRG tissues (L4/5 in rats and L3/4 in mice) using Trizol reagent (Invitrogen). Subsequently, 1000 ng of total RNA underwent reverse transcription with Takara reverse transcriptase (Japan). Quantitative PCR was performed using SYBR Green 2× PCR Master Mix (ThermoFisher Scientific) on an Applied Biosystems system (USA), with primer sequences detailed in Appendix Table S2. The PCR procedure included an initial denaturation step at 95 °C for 3 min, followed by 40 cycles of heat cycling at 95 °C for 10 s and 60 °C for 30 s. GAPDH was employed as the endogenous control for mRNA normalization. Following the completion of the cycles, melt curve analysis was conducted to confirm the absence of nonspecific products. Relative quantification was achieved using the comparative CT ($2^{-\Delta\Delta CT}$)

method. The specificity of the generated PCR product was confirmed by performing agarose gel electrophoresis and melting-curve analysis. The sequences of primers are listed in Appendix Table S2.

## Immunofluorescence

Following perfusion, the ipsilateral DRG was extracted and fixed overnight at 4 °C in 4% paraformaldehyde. DRG sections, with a thickness of 12–18 μm, were prepared using a Leica microtome (Germany). The sections were subjected to dehydration through sequential immersion in 10%, 20%, and 30% sucrose gradient solutions until complete sinking occurred. Immunofluorescence procedures were conducted in accordance with previously described methods. Concisely, the sections were first obstructed with 10% goat serum for a duration of 90 min at room temperature. Afterwards, they were subjected to overnight incubation at a temperature of 4 °C with the following primary antibodies: rabbit anti-GPR160 (1:200, Alomone Labs, AGR-090), mouse anti-NeuN (1:500, Novus, NBP1-92693), mouse anti-glutamine synthetase (GS, 1:400, Abcam, ab64613), mouse anti-CGRP (1:300, Sigma-Aldrich, C7113), isolectin B4 (IB4, 1:300, Sigma-Aldrich, L2140), mouse anti-NF200 (1:500, Sigma-Aldrich, N5389), mouse anti-H3K27me3 (1:300, Abcam, ab6002), mouse anti-H3K27ac (1:300, Thermo Scientific, MA5-42545), rabbit anti-Sp1 (1:200; Proteintech, 66508-1-Ig). To increase the specificity of the GPR160 antibody, the GPR160 antibody was preadsorbed against its blocking peptide (BLP-GR090, Alomone Labs) prior to incubation with DRG sections, by mixing the GPR160 antibody with the blocking peptide. The DAPI dye was used to label the nuclei of the cells. Secondary antibodies (Alexa Fluor® 488 donkey anti-rabbit, ab150073, Alexa Fluor® 594 donkey anti-rabbit, ab150076, Alexa Fluor® 488 donkey anti-mouse, ab150105, Alexa Fluor® 594 donkey anti-mouse, ab150108, 1:500) were administered for a period of 1 h at ambient temperature. Confocal analyses were conducted utilizing a Zeiss LSM880 confocal laser scanning microscope (Zeiss, Germany). Quantification of single-labeled cells was performed manually or with NIH ImageJ Software.

## RNAscope

Rats underwent deep anesthesia with isoflurane and were subjected to transcardial perfusion using PBS, followed by fixation with 4% paraformaldehyde. Subsequently, the L4/5 DRGs were extracted and underwent post-fixation in the same solution at 4 °C for 2 h. The tissues were then cryopreserved in a solution of 30% sucrose in PBS for a duration of 2 days. We acquired DRG slices with a thickness of 12 μm using a cryostat. The RNAscope equipment (Advanced Cell Diagnostics, USA) was used to perform in situ hybridization (ISH) following the manufacturer's procedure. The first procedure was removing moisture, followed by exposing the sample to hydrogen peroxide and protease IV at ambient temperature. Commercial probes targeting GPR160, Sp1, and NeuN were employed, following the Multiplex Fluorescent Kit v2 protocol. Following ISH, DRG sections were counterstained with DAPI (Vector Laboratories, USA). The capture of DRG images was achieved using a Zeiss LSM880 confocal laser scanning microscope (Zeiss, Germany).

## Chromatin immunoprecipitation-PCR

ChIP was conducted employing the SimpleChIP® Plus Enzymatic Chromatin IP Kit (Cell Signaling Technology) in accordance with the

manufacturer's directions. To summarize, DRG neurons from rats were obtained and placed in an ice-cold PBS solution that included a mixture of protease inhibitors. Protein-DNA crosslinking was accomplished through incubation with 1.5% formaldehyde, with glycine employed to quench the crosslinking reaction. Subsequently, chromatin underwent enzymatic digestion, and the resulting lysate was sonicated in a ChIP buffer to disrupt the nuclear membrane, yielding DNA fragments ranging from 200 to 1000 base pairs. Termination of the digestion process was achieved with the addition of 0.5 M EDTA. A 5% fraction of the supernatant was reserved as total chromatin input DNA, while the remaining supernatant was incubated with specific antibodies (H3K27me3 from Abcam-ab6002, H3K27ac from Abcam-ab4729, Sp1 from Proteintech-21962-1-AP) or an IgG antibody was employed as a negative control. This mixture was subjected to overnight rotation at 4 °C to facilitate chromatin binding. Immune complexes were captured using ChIP-grade protein G magnetic beads. Eluted DNA was subsequently purified, and samples were subjected to quantitative PCR analysis. The ChIP-enriched DNA was evaluated relative to normalized values obtained from the corresponding input sample. The PCR products were separated using 2% agarose gel electrophoresis.

## Luciferase reporter assays

The GPR160 gene promoter region's putative binding sites for the transcription factor Sp1 and CREB1 were predicted using the JASPAR (https://jaspar.genereg.net) and PROMO (http://alggen.lsi.upc.es) databases. To assess transcriptional activity, we conducted a luciferase reporter assay employing the pGL3-Promoter vector. GPR160 promoters were co-transfected with Sp1 or CREB1 overexpression plasmids into PC12 cells, facilitated by Lipofectamine 2000 (Invitrogen). PC12 cells were purchased from the cell bank of the Chinese Academy of Sciences (Shanghai, China). After 24 h, we quantified luciferase activity utilizing the Dual-Luciferase Reporter Assay System (Promega, E1910). The luciferase activity was determined by separately measuring firefly and Renilla luminescence, with firefly luminescence activity normalized to Renilla luciferase activity.

## Whole-cell patch-clamp recordings

Neurons were examined in an extracellular solution with the following composition (in mM): 140 NaCl, 3.5 KCl, 1 MgCl$_2$, 2 CaCl$_2$, 10 glucose, 10 HEPES, and 1.25 NaH$_2$PO$_4$. The solution's pH was adjusted to 7.4 using NaOH. Electrophysiological recordings were carried out using the EPC-10 USB amplifier (HEKA Electronics, Germany) in conjunction with the Patchmaster software (Heka, Germany) on a connected computer. Offset potentials were nullified immediately before establishing the seal, and no leakage subtraction was applied. Rapid capacitance compensation was enacted upon achieving a high-quality seal, followed by whole-cell capacitance compensation. The data was collected employing the current clamp mode with a digitization rate of 20 kHz. Patch clamp recordings were exclusively conducted on small-diameter DRG neurons with diameters less than 25 μm, corresponding to a membrane capacitance of less than 25 pF. Neurons with resting membrane potentials (RMP) exceeding −45 mV or falling below −80 mV were excluded from analysis. Series resistance was compensated to a value of ≤25 MΩ.

## Statistical analyses

The data is expressed as mean ± SEM and was analyzed using GraphPad Prism (version 9.0). The significance of differences between the two groups was measured employing a two-tailed Student's *t* test. For comparisons involving multiple groups or time points, one-way or two-way ANOVA with Tukey's post hoc tests was employed. A significance level of $P < 0.05$ was applied to all experiments.

## Data availability

mRNA microarray data has been deposited in NCBI GEO under accession code GSE270839 (https://www.ncbi.nlm.nih.gov/geo/query/acc.cgi?acc=GSE270839). Source data are provided with this paper.

The source data of this paper are collected in the following database record: biostudies:S-SCDT-10_1038-S44319-024-00292-6.

## Peer review information

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

## Acknowledgements

We thank Guangyin Xu (Soochow University, China) for manuscript proofreading. We thank Home for Researchers editorial team (www.home-for-researchers.com) for language editing service. This work was supported by the China Postdoctoral Science Foundation (2022M712310), National Natural Science Foundation of China (81901124, 82171216), Natural Science Foundation of Zhejiang Province (LGD22H090002), the Science and Technology Project of Jiaxing City (2023AY40022), the National Clinical Key Specialty Construction Project-Oncology department (2023-GJZK-001), the Zhejiang Provincial Traditional Chinese Medical Innovation Team of China under Grant No. 2022-19, the Key Medical Subjects Established by Zhejiang Province and Jiaxing City Jointly Pain Medicine and Oncology (2019-ssttyx, 2023-SSGJ-001), the Clinical Key Specialty of Zhejiang Province Anesthesiology (2023-ZJZK001), the Construction Project of Key Laboratory of Nerve and Pain Medicine in Jiaxing City, the Scientific research project of Anhui Provincial Health Commission (AHWJ2023A30069), the Clinical Key Specialty of Anhui Province Anesthesiology (2022-AH-105) and Bengbu Science and Technology Projects (20220115).

## Author contributions

**Chengfei Xu**: Resources; Data curation; Formal analysis; Funding acquisition; Methodology; Writing—original draft; Project administration; Writing—review and editing. **Yahui Wang**: Data curation; Formal analysis; Methodology; Writing—original draft. **Chaobo Ni**: Supervision; Validation; Writing—original draft. **Miao Xu**: Data curation; Methodology. **Chengyu Yin**: Data curation; Formal analysis; Methodology. **Qiuli He**: Data curation; Formal analysis. **Bing Ma**: Data curation; Formal analysis. **Jie Fu**: Supervision; Validation. **Baoxia Zhao**: Data curation; Formal analysis. **Liping Chen**: Data curation; Formal analysis. **Tong Zhi**: Data curation; Methodology. **Shirong Wei**: Supervision; Validation. **Liang Cheng**: Data curation; Formal analysis. **Hui Xu**: Formal analysis; Supervision; Validation. **Jiajun Xiao**: Data curation; Methodology. **Lei Yang**: Data curation; Formal analysis. **Qingqing Xu**: Data curation; Formal analysis. **Jiao Kuang**: Formal analysis; Methodology. **Boyi Liu**: Data curation; Formal analysis. **Qinghe Zhou**: Supervision; Validation; Methodology. **Xuewu Lin**: Conceptualization; Supervision; Validation; Methodology; Writing—original draft. **Ming Yao**: Data curation; Supervision; Funding acquisition; Validation; Writing—original draft; Writing—review and editing. **Huadong Ni**: Conceptualization; Resources; Data curation; Software; Formal analysis; Supervision; Funding acquisition; Validation; Investigation; Visualization; Methodology; Writing—original draft; Project administration; Writing—review and editing.

Source data underlying figure panels in this paper may have individual authorship assigned. Where available, figure panel/source data authorship is listed in the following database record: biostudies:S-SCDT-10_1038-S44319-024-00292-6.

## Disclosure and competing interests statement

The authors declare no competing interests.

# Expanded View Figures

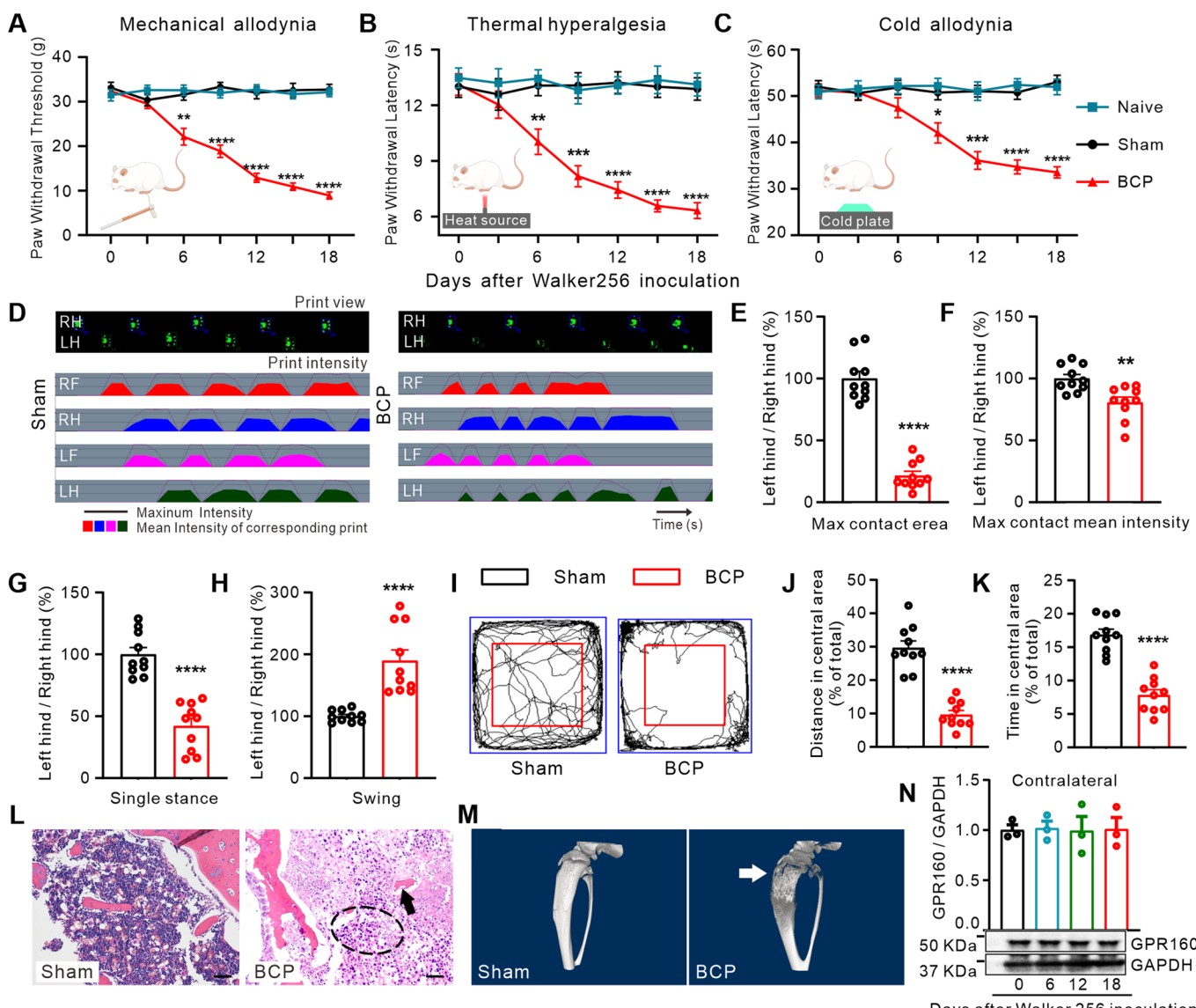

**Figure EV1. Tumor infiltration to establish a bone cancer pain model induces pain-related behaviors in rats.**

(A–C) Paw withdrawal threshold (PWT) to von Frey filament stimuli (A, **$P = 0.0019$, ****$P < 0.0001$ versus sham group) as well as paw withdrawal latencies (PWLs) to thermal (B, **$P = 0.0081$, ***$P = 0.0001$, ****$P < 0.0001$ versus sham group) and cold (C, *$P = 0.0127$, ****$P < 0.0001$ versus sham group) stimuli on the ipsilateral (Ipsi) at various time points (days 0, 3, 6, 9, 12, 15 and 18) after BCP or sham surgery. Data are mean ± SEM of biological replicates $n = 10$ rats/group, two-way ANOVA with repeated measures followed by the post hoc Tukey test. (D) Representative CatWalk gait, including print view and print intensity, for sham and BCP group. (E–H) Tumor infiltration resulted in a reduction of maximum contact area (E, ****$P < 0.0001$ versus sham group), maximum contact mean intensity (F, **$P = 0.0024$ versus sham group), and single stance duration (G, ****$P < 0.0001$ versus sham group), along with an increase in swing duration (H, ****$P < 0.0001$ versus sham group) among tumor-bearing rats. LH, left hind; RH, right hind. Student's unpaired $t$ test, Data are mean ± SEM of biological replicates $n = 10$ rats. (I) Representative rodent tracks recorded from rats 12 days post-surgery during the open field test. (J, K) Center distance (J, ****$P < 0.0001$ versus sham group) and time (K, ****$P < 0.0001$ versus sham group) for rats 12 days post-surgery in the open field test. Student's unpaired $t$ test, Data are mean ± SEM of biological replicates $n = 10$ rats. (L) Hematoxylin and eosin (H&E) staining revealed the infiltration of tumor cells, accompanied by medullary bone loss and tibial bone destruction, on POD 18. Scale bar: 50 μm. (M) Representative micro-CT images showing tibia bone microstructure in sham and BCP rats on POD 18. (N) Representative immunoblots and summarized data showing the GPR160 protein expression in the contralateral L4/5 DRG on days 0, 6, 12, and 18 post-BCP. One-way ANOVA with repeated measures followed by post hoc Tukey test, Data are mean ± SEM of biological replicates $n = 3$ rats/group, $P = 0.9993$ versus sham group. Source data are available online for this figure.

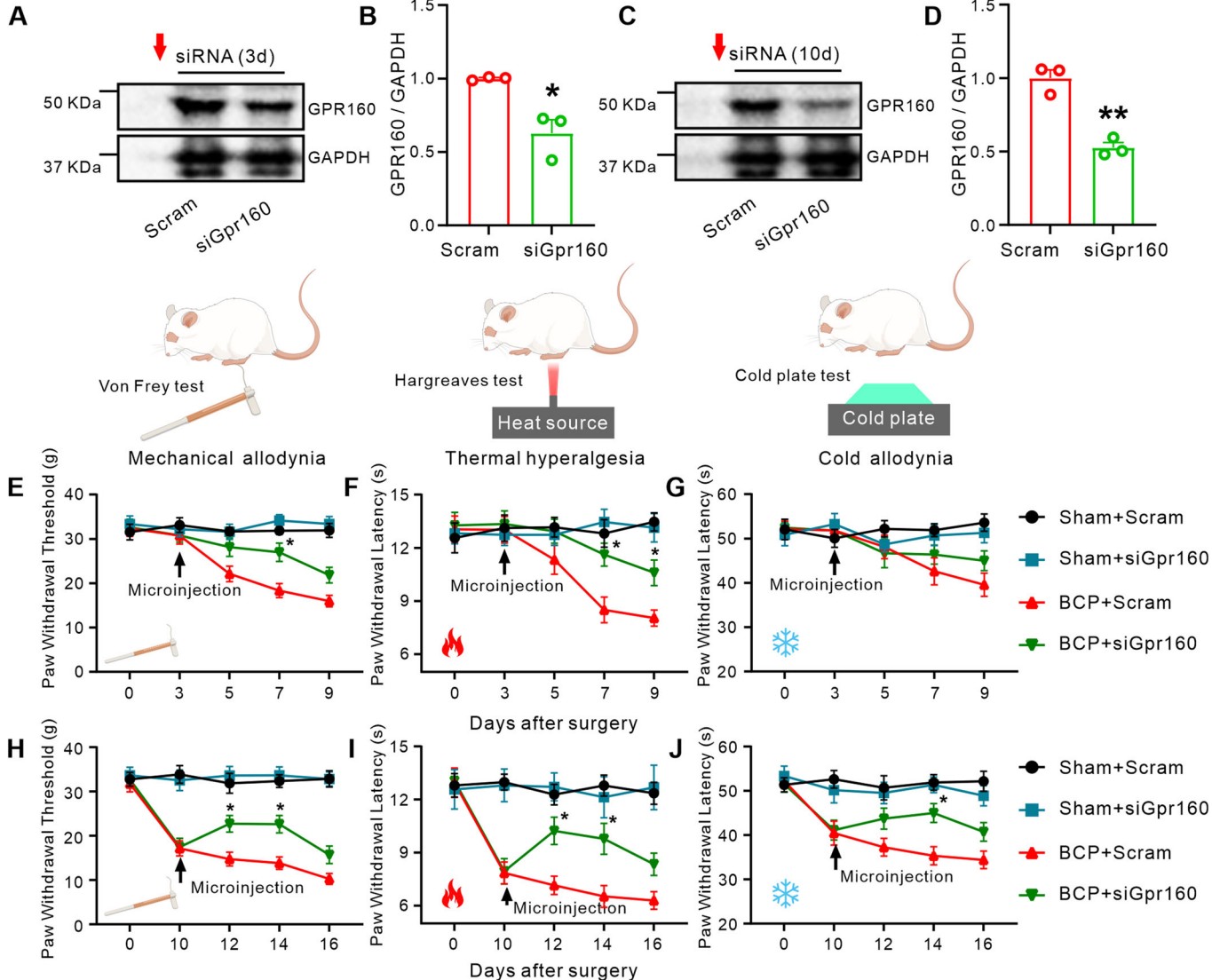

**Figure EV2. DRG increased GPR160 is required for development and maintenance of BCP in rats.**

(A–D) Effect of microinjection of *Gpr160* siRNA (*siGpr160*) or scramble siRNA (Scram) into the ipsilateral L4/5 DRG in rats on the expression of GPR160 on POD 7 (B, *P = 0.0195 versus BCP plus Scram group) and POD 14 (D, **P = 0.0016 versus BCP plus Scram group). Student's unpaired *t* test. Data are mean ± SEM of biological replicates n = 3 rats/group. The red arrow represents microinjection. (E–G) Effect of pre-microinjection of *siGpr160* or Scram into the ipsilateral L4/5 DRG of rats on the development of BCP-induced mechanical allodynia (E, *P = 0.0251 versus BCP plus Scram group) and heat hyperalgesia (F, *P = 0.0317, *P = 0.0496 versus BCP plus Scram group), with no discernible effect on cold allodynia (G, P = 0.4244 versus BCP plus Scram group) on the ipsilateral side. Data are mean ± SEM of biological replicates n = 7–8 rats/group. Two-way ANOVA with repeated measures followed by post hoc Tukey test. (H–J) Effect of siRNA microinjection of siGpr160 or Scram into the ipsilateral L4/5 DRG of rats on the persistence of BCP-induced mechanical allodynia (H, *P = 0.0302, *P = 0.0171 versus BCP plus Scram group), heat hyperalgesia (I, *P = 0.0302, *P = 0.0472 versus BCP plus Scram group) and cold allodynia (J, *P = 0.0282 versus BCP plus Scram group) on the ipsilateral side. Data are mean ± SEM of biological replicates n = 7–8 rats/group. Two-way ANOVA with repeated measures followed by post hoc Tukey test. Source data are available online for this figure.

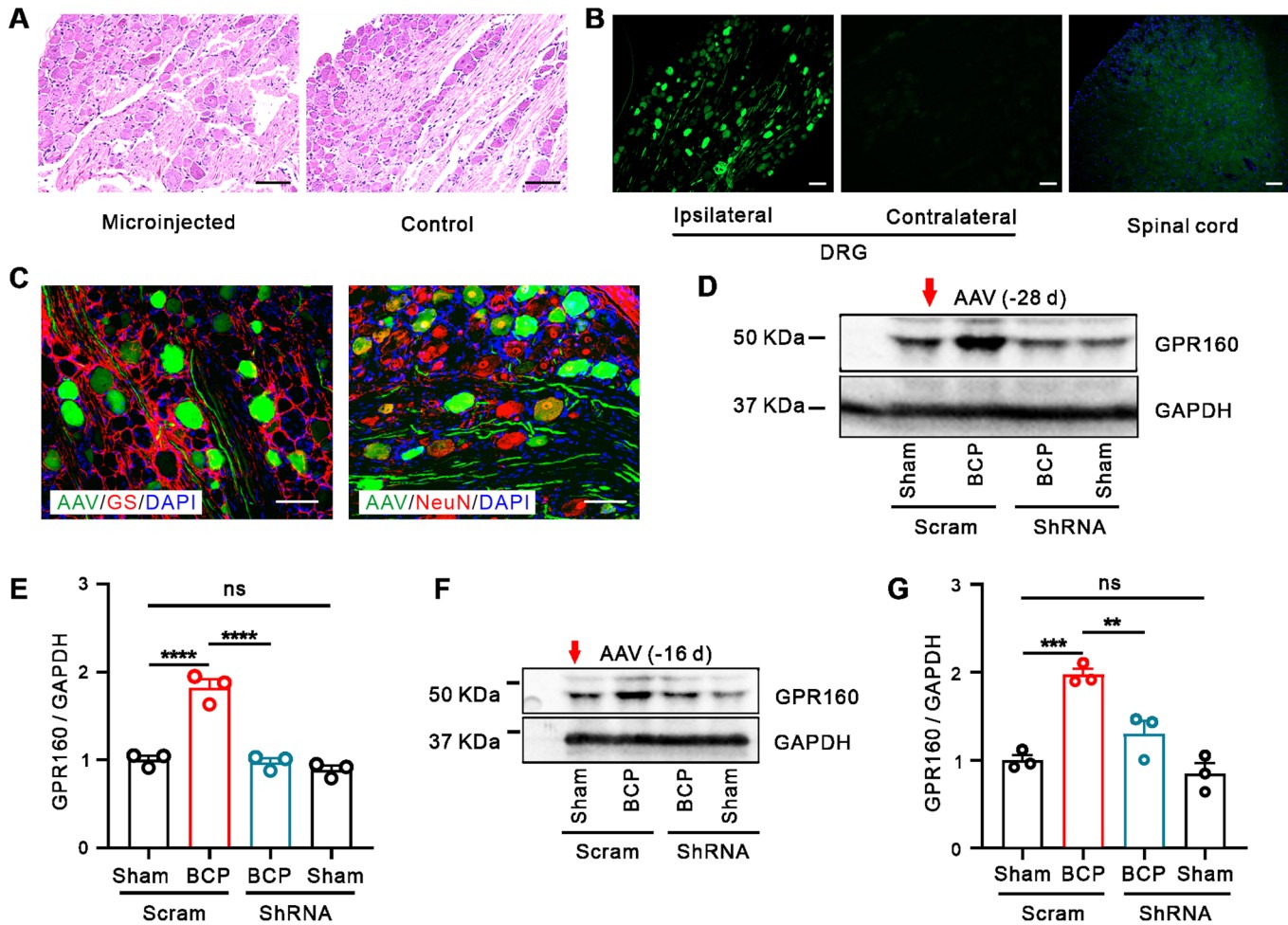

**Figure EV3. Validation of targeted gene delivery and expression.**

(A) Representative hematoxylin and eosin-stained images of control un-injected and injected DRGs, captured 28 days post-microinjection. Scale bar: 100 μm. (B) Representative images displaying GFP expression in the ipsilateral and contralateral L4/5 DRG and spinal cord at 28 days post-microinjection with AAV-*Gpr160* (*Gpr160*) or AAV-*Gfp* (*Gfp*) into the ipsilateral L4/5 DRG. *n* = 3 rats. Scale bar: 100 μm. (C) Representative images of co-localization of GFP and GS, or NeuN in the AAV-injected DRG. *n* = 3 rats. Scale bar: 100 μm. (D–G) Representative immunoblots (D, F) and summarized data (E, ****P < 0.0001 versus the BCP plus Scram group, P = 0.8416 versus the Sham plus Scram group and G, ***P = 0.0008, **P = 0.0085 versus the BCP plus Scram group, P = 0.6486 versus the Sham plus Scram group) illustrating the GPR160 protein expression on POD 18 in the ipsilateral L4/5 DRG pre-microinjected with AAV-*Gpr160* (*Gpr160*) or control AAV-*Gfp* (*Gfp*) 28 days (D, E) and 16 days (F, G) prior to surgery. Two-way ANOVA with repeated measures followed by post hoc Tukey test, Data are mean ± SEM of biological replicates *n* = 3 rats/group. The red arrow represents microinjection. Source data are available online for this figure.

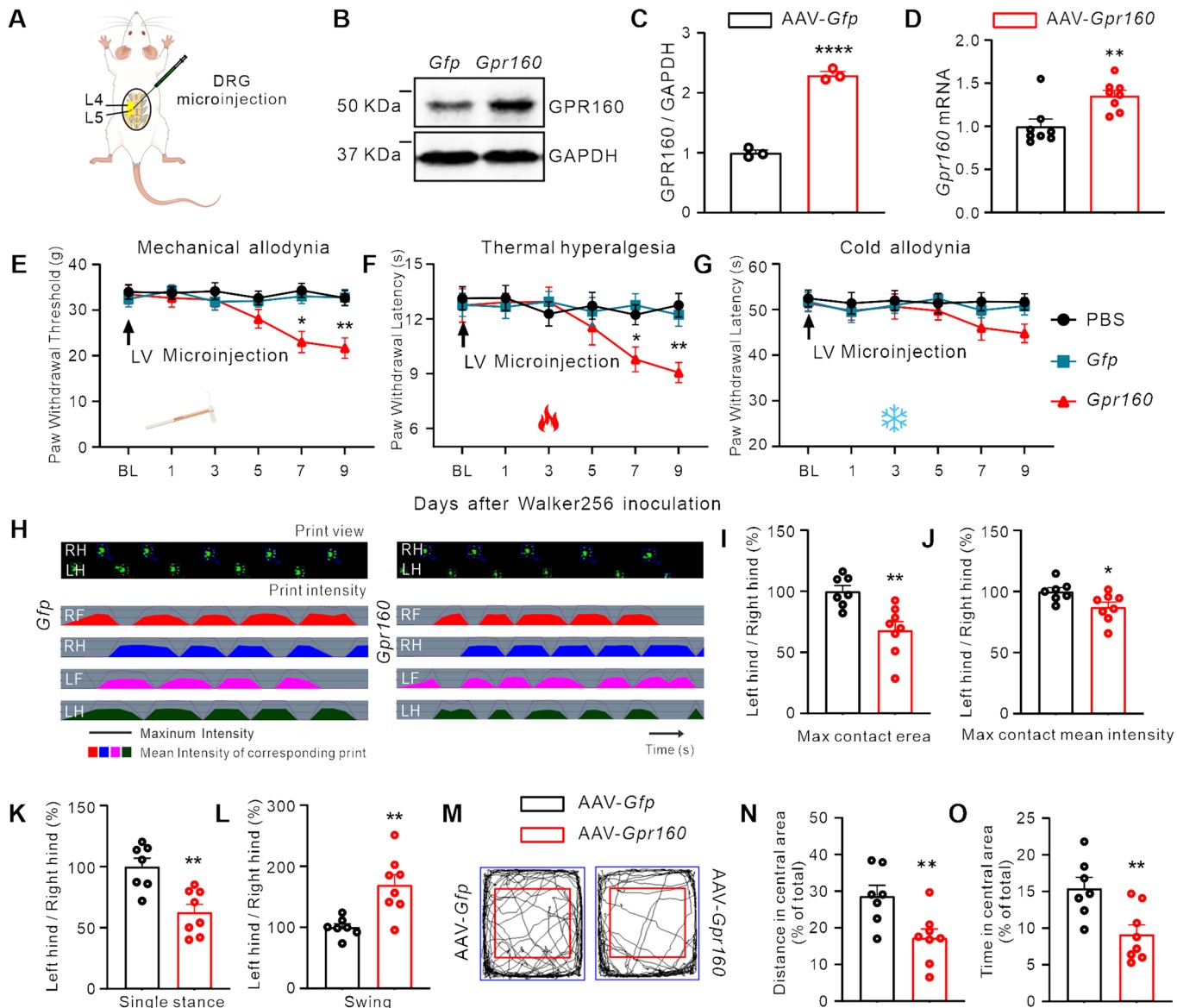

**Figure EV4.   Effect of DRG GPR160 overexpression on nociceptive thresholds and pain-related behavior in naive female rats.**

(**A**) Schematic protocol for DRG microinjection in rat. (**B**, **C**) Representative immunoblots and summarized data showing GPR160 protein expression in the ipsilateral L4/5 DRG of rats at 5 weeks post-microinjection with AAV-*Gpr160* (*Gpr160*) or AAV-*Gfp* (*Gfp*). Student's unpaired t test, Data are mean ± SEM of biological replicates n = 3 rats/group, ****P < 0.0001 versus AAV-*Gfp* group. (**D**) The *Gpr160* mRNA expression in the ipsilateral DRG of rats at 5 weeks post-microinjection with AAV-*Gpr160* or AAV-*Gfp*. Student's unpaired t test, Data are mean ± SEM of biological replicates n = 8 rats/group, **P = 0.0041 versus AAV-*Gfp* group. (**E–G**) Paw withdrawal responses to mechanical (**E**, *P = 0.0114, **P = 0.0058 versus the *Gfp* group), heat (**F**, *P = 0.0193, **P = 0.0055 versus the *Gfp* group), and cold (**G**, P = 0.1229 versus the *Gfp* group) stimuli on the ipsilateral side, recorded at specified time points following microinjection of LV-*Gpr160* (*Gpr160*), LV-*Gfp* (*Gfp*), or PBS into unilateral L4/5 DRGs in naive rats. Data are mean ± SEM of biological replicates n = 7–8 rats/group. Two-way ANOVA with repeated measures followed by post hoc Tukey test. (**H**) Representative CatWalk gait, including print view and print intensity, within the DRG microinjection groups treated with AAV-*Gfp* or AAV-*Gpr160*. (**I–L**) DRG microinjection of AAV resulted in a reduction of max contact area (**I**, **P = 0.0032 versus AAV-*Gfp* group), max contact mean intensity (**J**, *P = 0.0309 versus AAV-*Gfp* group) and single stance (**K**, **P = 0.0017 versus AAV-*Gfp* group), along with an increase in swing duration (**L**, **P = 0.0027 versus AAV-*Gfp* group) in naive female rats. Student's unpaired t test, Data are mean ± SEM of biological replicates n = 7–8 rats/group. (**M**) Representative animal tracks observed in naive rats following AAV microinjection during the open field test. (**N**, **O**) Center time (**N**, **P = 0.0071 versus AAV-*Gfp* group) and distance (**O**, **P = 0.0098 versus AAV-*Gfp* group) measurements for rats 5 weeks post-AAV microinjection during the open field test. Student's unpaired t test, Data are mean ± SEM of biological replicates n = 7–8 rats/group. Source data are available online for this figure.

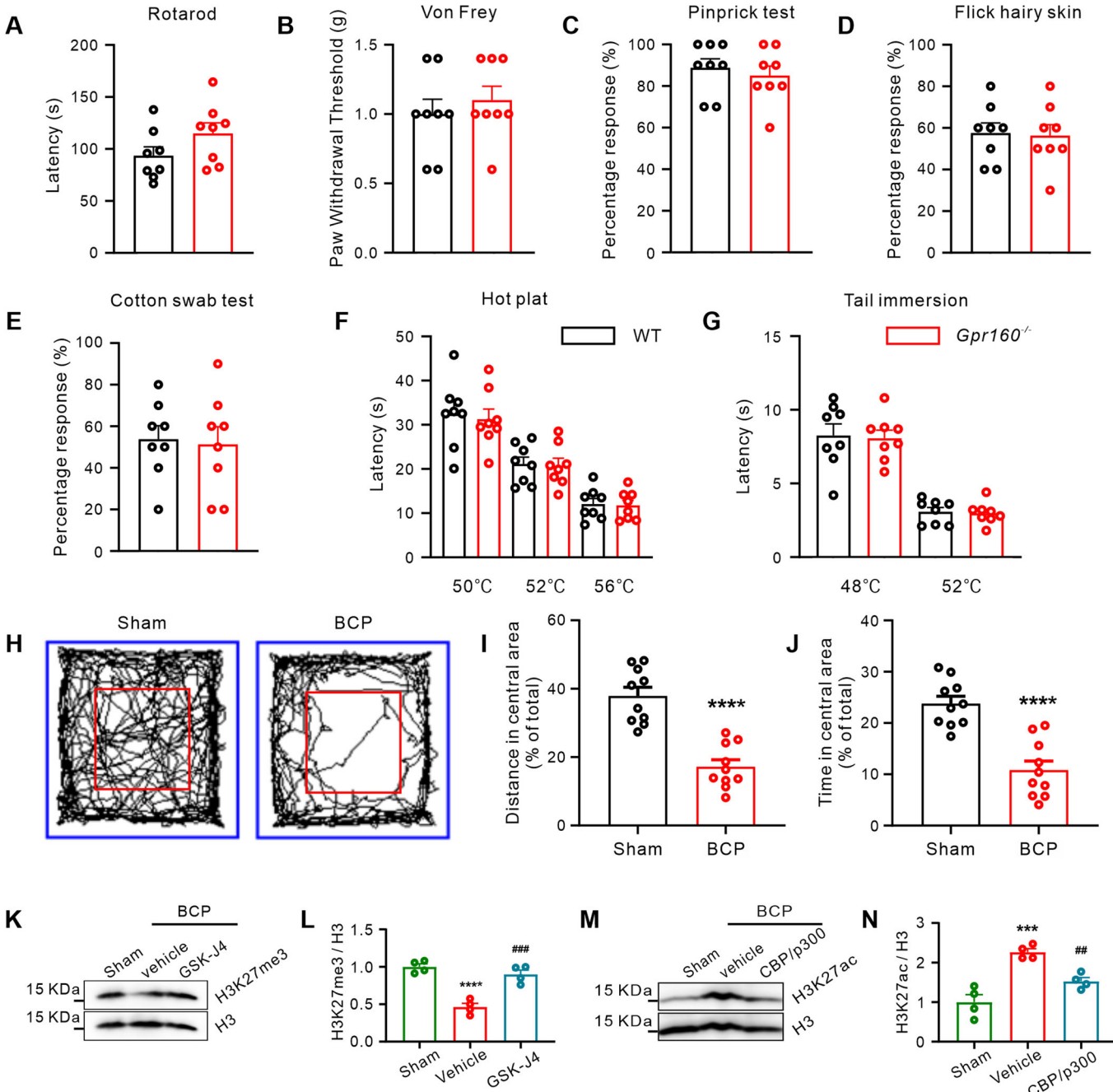

**Figure EV5. Basal motor and sensory function in WT and *Gpr160*^-/- mice, and establishment of the mouse bone cancer pain model.**

(A) Motor function was tested through Rotarod assay. Student's unpaired *t* test, Data are mean ± SEM of biological replicates n = 8 mice/group, *P* = 0.1301 versus WT group. (B, C) Mechanical pain sensation was measured through von Frey test (B, *P* = 0.5057 versus WT group) and pinprick test (C, *P* = 0.5667 versus WT group). Student's unpaired *t* test, Data are mean ± SEM of biological replicates *n* = 8 mice/group. (D, E) Evaluation of light-touch sensation was conducted via the flick hairy skin test (D, *P* = 0.8654 versus WT group) and cotton swab test (E, *P* = 0.8195 versus WT group). Student's unpaired *t* test, Data are mean ± SEM of biological replicates n = 8 mice/group. (F, G) Thermal sensation was gauged using the hot plate assay (F, *P* = 0.7378, *P* = 0.8807, *P* = 0.8572 versus WT group) and tail immersion assay (G, *P* = 0.8463, *P* = 0.8006 versus WT group). Student's unpaired *t* test, Data are mean ± SEM of biological replicates *n* = 8 mice/group. (H) Representative animal tracks were recorded from mice 18 d after surgery in the open field test. (I, J) Measurements of distance (I, ****P < 0.0001 versus WT group) and time (J, ****P < 0.0001 versus WT group) in the center were obtained for mice 12 d post-surgery in the open field test. Student's unpaired *t* test, Data are mean ± SEM of biological replicates *n* = 10 mice/group. (K, L) Representative immunoblots (K) and summarized data (L) showing the decreased protein expression level of H3K27me3 in DRG on POD 15 after BCP was intensified by GSK-J4. One-way ANOVA with repeated measures followed by post hoc Tukey test, Data are mean ± SEM of biological replicates *n* = 4 rats/group, ****P < 0.0001 versus Sham group. ###*P* = 0.0004 versus BCP plus vehicle group. (M, N) Representative immunoblots (M) and summarized data (N) showing the increased protein expression level of H3K27ac in DRG on POD 15 after BCP was attenuated by CBP/p300. One-way ANOVA with repeated measures followed by post hoc Tukey test, Data are mean ± SEM of biological replicates *n* = 4 rats/group, ***P = 0.0002 versus Sham group. ##*P* = 0.0091 versus BCP plus vehicle group. Source data are available online for this figure.

