## [Peer Review File · EMBO Reports]

Histone modifications and Sp1 promote GPR160 expression in bone cancer pain within rodent models

Chengfei Xu, Yahui Wang, Chaobo Ni, Miao Xu, Chengyu Yin, Qiuli He, Bing Ma, Jie Fu, Baoxia Zhao, Liping Chen, Tong Zhi, Shirong Wei, Liang Cheng, Hui Xu, Jiajun Xiao, Lei Yang, Qingqing Xu, Jiao Kuang, Boyi Liu, Qinghe Zhou, Xuewu Lin, Ming Yao, and Huadong Ni

Corresponding author(s): Huadong Ni (huadongni@zjxu.edu.cn) , Xuewu Lin (linxuewu@bbmc.edu.cn), Ming Yao (jxyaoming@zjxu.edu.cn)

Review Timeline:

Submission Date:	28th Mar 24
Editorial Decision:	13th May 24
Revision Received:	10th Aug 24
Editorial Decision:	12th Sep 24
Revision Received:	15th Sep 24
Accepted:	1st Oct 24

Transaction Report:

Dear Prof. Ni

Thank you for the submission of your research manuscript to our journal. We have now received the full set of referee reports that is copied below.

As you will see, the referees acknowledge that the findings are interesting and that the conclusions on Gpr160 in bone cancer pain are overall supported by the data presented. That said, a major concern raised by all three referees is the rather weak link between Gpr160 and TRPV1. Upon further discussion, all three referees supported the suggestion from referee #3, to focus your study on the role of Gpr160 in bone cancer pain and its regulation by epigenetic factors. The data on TRPV1 should be removed and the mechanism/ion channel involvement might be the focus of a follow-up study.

Given these constructive and supportive comments, I would like to invite you to revise your study along the lines suggested above for potential publication in EMBO Reports. Please also address all other concerns regarding missing information in the methods and unclear description of experimental details.

Please also address all referee concerns in a complete point-by-point response. Acceptance of the manuscript will depend on a positive outcome of a second round of review. It is EMBO Reports policy to allow a single round of revision only and acceptance or rejection of the manuscript will therefore depend on the completeness of your responses included in the next, final version of the manuscript.

We realize that it is difficult to revise to a specific deadline. In the interest of protecting the conceptual advance provided by the work, we recommend a revision within 3 months (August 13th). Please discuss the revision progress ahead of this time with the editor if you require more time to complete the revisions.

I am also happy to discuss the revision further via e-mail or a video call, if you wish.

*******IMPORTANT NOTE:**

We perform an initial quality control of all revised manuscripts before re-review. Your manuscript will FAIL this control and the handling will be delayed IN CASE the following APPLIES:

- 1) A data availability section providing access to data deposited in public databases is missing. If you have not deposited any data, please add a sentence to the data availability section that explains that.
- 2) Your manuscript contains statistics and error bars based on $n=2$. Please use scatter blots in these cases. No statistics should be calculated if $n=2$.

When submitting your revised manuscript, please carefully review the instructions that follow below. Failure to include requested items will delay the evaluation of your revision. *****

- 1) a .docx formatted version of the manuscript text (including legends for main figures, EV figures and tables). Please make sure that the changes are highlighted to be clearly visible.
- 2) individual production quality figure files as .eps, .tif, .jpg (one file per figure). Please download our Figure Preparation Guidelines (figure preparation pdf) from our Author Guidelines pages <https://www.embopress.org/page/journal/14693178/authorguide> for more info on how to prepare your figures.
- 3) a .docx formatted letter INCLUDING the reviewers' reports and your detailed point-by-point responses to their comments. As part of the EMBO Press transparent editorial process, the point-by-point response is part of the Review Process File (RPF), which will be published alongside your paper.
- 4) a complete author checklist, which you can download from our author guidelines (<<https://www.embopress.org/page/journal/14693178/authorguide>>). Please insert information in the checklist that is also reflected in the manuscript. The completed author checklist will also be part of the RPF.
- 5) Please note that all corresponding authors are required to supply an ORCID ID for their name upon submission of a revised manuscript (<<https://orcid.org/>>). Please find instructions on how to link your ORCID ID to your account in our manuscript tracking system in our Author guidelines (<<https://www.embopress.org/page/journal/14693178/authorguide#authorshipguidelines>>)

6) We replaced Supplementary Information with Expanded View (EV) Figures and Tables that are collapsible/expandable online. A maximum of 5 EV Figures can be typeset. EV Figures should be cited as 'Figure EV1, Figure EV2' etc... in the text and their respective legends should be included in the main text after the legends of regular figures.

7) Please note that a Data Availability section at the end of Materials and Methods is now mandatory. In case you have no data that requires deposition in a public database, please state so instead of refereeing to the database. Please do not refer to data "available on request" in this section.

See also < <https://www.embopress.org/page/journal/14693178/authorguide#dataavailability>>. Please note that the Data Availability Section is restricted to new primary data that are part of this study.

Additional information on source data and instruction on how to label the files are available <<https://www.embopress.org/page/journal/14693178/authorguide#sourcedata>>.

10) Figure legends and data quantification:

- the name of the statistical test used to generate error bars and P values,
 - the number (n) of independent experiments (please specify technical or biological replicates) underlying each data point,
 - the nature of the bars and error bars (s.d., s.e.m.)
- If the data are obtained from n {less than or equal to} 5, show the individual data points in addition to the SD or SEM.
- If the data are obtained from n {less than or equal to} 2, use scatter blots showing the individual data points.

11) Our journal encourages inclusion of *data citations in the reference list* to directly cite datasets that were re-used and obtained from public databases. Data citations in the article text are distinct from normal bibliographical citations and should directly link to the database records from which the data can be accessed. In the main text, data citations are formatted as follows: "Data ref: Smith et al, 2001" or "Data ref: NCBI Sequence Read Archive PRJNA342805, 2017". In the Reference list, data citations must be labeled with "[DATASET]". A data reference must provide the database name, accession number/identifiers and a resolvable link to the landing page from which the data can be accessed at the end of the reference. Further instructions are available at <<https://www.embopress.org/page/journal/14693178/authorguide#referencesformat>>.

12) All Materials and Methods need to be described in the main text. We would encourage you to use 'Structured Methods', our new Methods format. According to this format, the Methods section should include a Reagents and Tools Table (listing key reagents, experimental models, software and relevant equipment and including their sources and relevant identifiers) followed by a Methods and Protocols section in which we encourage the authors to describe their methods using a step-by-step protocol format with bullet points, to facilitate the adoption of the methodologies across labs. More information on how to adhere to this format as well as downloadable templates (.doc or .xls) for the Reagents and Tools Table can be found in our author guidelines: < <https://www.embopress.org/page/journal/14693178/authorguide#manuscriptpreparation>>.

An example of a Method paper with Structured Methods can be found here:
<<https://www.embopress.org/doi/10.15252/msb.20178071>>.

13) As part of the EMBO publication's Transparent Editorial Process, EMBO Reports publishes online a Review Process File to accompany accepted manuscripts. This File will be published in conjunction with your paper and will include the referee reports, your point-by-point response and all pertinent correspondence relating to the manuscript.

I look forward to seeing a revised form of your manuscript when it is ready.

Yours sincerely,

Referee #1:

In this manuscript by Xu et al, the authors evaluate the role for Gpr160 in a model of bone cancer induced pain. The results demonstrated that Gpr160 mRNA and protein levels were increased in the DRGs of animals experiencing CIBP and knockdown of Gpr160 prevented behavioral effects of CIBP. The proposed mechanism of Gpr160 on modulation of pain is increased functional response of TRPV1.

Strengths:

1. Although the role for Gpr160 in other pain conditions have been proposed by Dr. Salvemini's group, its contribution in CIBP is unknown.
2. The behavioral characterization of Gpr160 role in CIBP is thoroughly performed including AAV mediated knockdown, germline KO and use of two species.

However, following concerns dampen my enthusiasm somewhat.

Major:

1. The implication that only TRPV1 is mediating Gpr160 effects are not well supported. E.g. Fig 4 shows change in AP and current thresholds in the Gpr160 knockdown animals which suggests channels other than TRPV1 that underlie threshold setting are affected. Although the data in Fig 8 point towards increased TRPV1 function in CIBP DRG neurons that is dependent on presence of Gpr160, how presence or activation of Gpr160 leads to enhanced TRPV1 function remains unanswered.
2. It appears to me that TRPM8 levels are also changed in CIBP DRG neurons, there was no further attempt to explore that.
3. The data in Fig 8E-G suggests that capsaicin behavioral response is dependent on the presence of Gpr160. This is hard to explain given their own data that TRPV1 expression is not affected by absence of Gpr160. Are the authors suggesting that capsaicin does not bind to TRPV1 or TRPV1 is not present on plasma membrane in absence of Gpr160?
4. Gpr160 overexpression model could be utilized to identify pathways by which TRPV1 activity is enhanced to consolidate the hypothesis.
5. The concentration of capsaicin used in Fig 8 calcium imaging experiments is 500 micromolar. This is orders of magnitude above the typical concentrations of capsaicin used in these experiments by many other groups. Is this a typo?

Minor:

1. Fig 1: It would be useful to show separate marker images along with total neuronal count before the merged image (N and O) so that the level of co-expression can be assessed better. Inclusion of a colocalization coefficient analysis would also be helpful (also applies to other IHC figures).
2. Fig 3: It would help the reader if description of the groups (i.e. sham/sham+sh etc.) were listed together rather than on

different panels.

3. Fig 5: A and C: The western blots do not have the protein lanes well separated making quantification difficult.

Referee #2:

Authors Xu et al., explored the role of orphan GPCR GPR160 in bone cancer pain (BCP) modelled using multiple in vivo rodent models, specifically relating to hyperexcitability in small diameter neurons in the DRG. A large number of behavioral tests, along with transgenic mice, siRNA and AAV tools, and patch clamp showed a key role for Gpr160 in the development of hypersensitivity. DNA, RNA and protein variability and expression were assessed, to relate an increase in Gpr160 mRNA and protein in the DRG to increased binding of transcription factor Sp1 driven by histone modification (reduced H3K27me3 and increased H3K27ac). Increased GPR160 was subsequently associated with increased TRPV1 function in the DRG using calcium imaging and behavior. In this manuscript, multiple techniques are used to comprehensively characterize the role of H3K27me3/H3K27ac-SP1-GPR160 in the development of hypersensitivity in both mice and rats. The link to increased TRPV1 function is found to be weaker, despite this, the manuscript comprehensively characterizes an interesting and understudied pathway. However, a number of major and minor points should be addressed:

1. While concordant results from multiple models are compelling (despite the cancer-cell and tibia/femur bone variation), the sheer volume of data presented makes it difficult to easily distinguish what the source of the data is. An effort should be made to differentiate more clearly both in the methods and in the results. Adding timelines for the use of different techniques in each model, as well as for cell-culture experiments would aid the legibility. Additionally, reorganization of methods is recommended to reflect whether a technique was used for mice or rats, or even primary cultures.
2. The genome-wide mRNA expression profiling analysis in figure 1D, E, F is unclear, what is the source of this data? Adding details of the design of this experiment (n's, methods, statistical analysis etc) would aid in the interpretation of the results.
3. Briefly, experiments with an additional rat model (MADB-breast cancer cells) are presented and sex-differentiation is addressed in supp F2 with this model. Addition of stats of sex-matching/differentiation in the other models would be valuable. In contrast, the addition of results from the MADB-model are not found to add significant value to the story at this time, perhaps exclusion to prioritize clarification of other data would be beneficial.
4. Experiments with a PC12 cell line (co-immunoprecipitation, luciferase reporter assay) are described, however, the source of cell line is not mentioned in the methods.
5. The tests used to collect behavioral results, particularly presented in figure 3 and figure 8, are unclear in the current format, especially considering the large volume of behavioral tests described in the methods. A redesign of the figure is recommended to aid legibility.
6. In the presentation of calcium imaging results in figure 8, there appears to be a discrepancy between the KCl concentration indicated in the legend (40mM) and the methods (50mM). Additionally, a 500uM concentration of capsaicin sounds disproportionately high, perhaps this is an error? Additionally, in Figure 8E-G the capsaicin injection is given as 10ug, and the injection volume or timeline of experiment are not characterized in the methods.
7. In figure 4 the effect of BCP on excitability of small-diameter neurons is assessed with patch, Gpr160 KO is associated with decreased Ap, however, n-number is unclear, are these neurons from 1 animal? Also, a change is noted in the rheobase between cells from Wt/KO-BCP mice and AAV-injected (for over-expression of Gpr160) mice, perhaps this can be addressed in the results?
8. While the mechanism of Gpr160 is well characterized, the link to TRPV1 function is less convincing at this time. Co-expression of Gpr160 and TRPV1 in small-diameter neurons, as well as BCP affecting TRPV1 function are expected results. Lowered capsaicin response in Gpr160 KO cells and mice may be due to uncharacterized interactions, rather than direct effect of Gpr160 on TRPV1. Perhaps the title/introduction could be changed to refrain from predicting a direct correlation to TRPV1 function.

Referee #3:

"Histone modification promotes GPR160 contributes to bone cancer pain via regulating TRPV1 in rodents " reports novel results derived from insightful experiments with a topic of pathogenesis of bone cancer pain. - One could have said "... with a FOCUS on pathogenesis of bone cancer pain." But that is exactly the central weakness of the study, namely it is not strong on FOCUS. On the other hand, the paper, in many of its sections, shows rigor and depth so that data do indeed support the authors' claims.

As alluded in the opener, my most important issue with this paper is that it represents a piece of work - a lot of hard work - that hurts itself by being overboarding, in a way overloading the plate and thus detracting from its value.

If we start with the title: "Histone modification promotes GPR160 contributes to bone cancer pain via regulating TRPV1 in rodents" - did the authors mean "Histone modification enhances GPR160 expression and function as an important contributor to bone cancer pain in rodents" ? And here is my point: this could be a suitable title, but then "Histone modification enhances GPR160 expression and function as an important contributor to bone cancer pain in rodents, with suggested co-contribution of upregulated TRPV1 function" clearly is odd and shows the dilemma of having generated an enormous amount of data and trying to force them under one headline.

Having read the paper carefully and repeatedly - which also made clear, more and more with every re-read, that the paper does not read well, likely driven by the overabundance of what authors aim to report: My take-home is that core elements of the study are the regulated expression of GPR160 in bone cancer pain in rodents, epigenetic mechanisms of Gpr160 gene regulation and in-vivo role of GPR160, the latter being studied using gene knockdown and knockout and also - it is laudable and bespeaking of the authors' diligence - gain-of-function studies.

It is my recommendation that the study should wrap these important findings up and explore the related TRPV1 mechanisms separately.

Demonstrated altered electrical excitability of DRG neurons is rather linked directly to endogenous excitability machinery (voltage gated channels, K⁺ channels) than to TRP channels. Perhaps the authors can explore more of the mechanistic link between GPR160 and TRPV1, such as involvement of cell-autonomous signaling mechanisms involving phospholipids, lipases, kinases etc. and structural determinants in the GPR and the TRP channel. Also, whether this is a DRG mechanism that is coupled to the previously demonstrated co-function TRPV1-ANO1 (thus linking calcium influx with chloride eflux), which powerfully amplifies noci-transduction in DRG neurons.

Beyond careful proofing and editing for proper use of English the authors need to shorten their Discussion to enhance readability.

However, what needs to be included in the revised Discussion, guiding perhaps future experimentation, are references to known bone cancer pain mechanisms in clinics, and also more recent bone cancer pain mechanisms that are more directly coupled to mechanosensory mechanisms. Re the latter, TRPV1 (and GPR160) are not strongly supported as involved in mechanical sensing/ mechanotransduction.

The clinical evidence is clear that NGF signaling and mu-opioid receptor signaling are critically involved in human bone cancer pain. - What could be the linke to the authors' new findings ?

More recent preclinical studies suggest involvement of intrinsically mechanically sensitive signaling mechanisms in bone cancer pain, these should be discussed.

Finally, three specific details appear worthy of comment:

1) The authors state "Given that the initial pathological alterations in DRG serve as the primary instigators of chronic pain, ..." The fact is that there are virtually certainly other primary instigators of chronic pain OUTSIDE the DRG.

2) Fig. 1 - antibody validation needs to be conducted with Gpr160 knockout mouse tissue, such a finding conveys higher rigor than blocking of antibody binding with immunizing peptide.

3) Define the AAV2/9 capsid better - AAV9 typically has been used for targeting neural substrate, which has now been replaced with the PHP.S capsid (a AAV9 derivative) when targeting DRG neurons.

Responses to reviewers

Senior Editor

As you will see, the referees acknowledge that the findings are interesting and that the conclusions on Gpr160 in bone cancer pain are overall supported by the data presented. That said, a major concern raised by all three referees is the rather weak link between Gpr160 and TRPV1. Upon further discussion, all three referees supported the suggestion from referee #3, to focus your study on the role of Gpr160 in bone cancer pain and its regulation by epigenetic factors. The data on TRPV1 should be removed and the mechanism/ion channel involvement might be the focus of a follow-up study.

Dear editors and reviewers,

Thank you for your letter and for the reviewers' comments on our manuscript entitled "Histone modification promotes peripheral sensory neurons GPR160 transcription and contributes to BCP via regulating TRPV1 function in rodents" (ID: EMBOR-2024-59294V1). All of these comments were very helpful for revising and improving our paper. We have studied these comments carefully and have made corresponding corrections that we hope will meet with your approval. All three referees suggested to focus our study on the role of GPR160 in bone cancer pain and its regulation by epigenetic factors. Therefore, the mechanism/ion channel involvement in BCP will be the focus in our future experiments and we change the title of our article to " Histone modification enhances GPR160 expression and function as an important contributor to bone cancer pain in rodents ". The changes in the revised manuscript are highlighted in red. The responses to the reviewers' comments are provided below.

Referee #1:

In this manuscript by Xu et al, the authors evaluate the role for Gpr160 in a model of bone cancer induced pain. The results demonstrated that Gpr160 mRNA and protein levels were increased in the DRGs of animals experiencing CIBP and knockdown of Gpr160 prevented behavioral effects of CIBP. The proposed mechanism of Gpr160 on modulation of pain is increased functional response of TRPV1.

Strengths:

1. Although the role for Gpr160 in other pain conditions have been proposed by Dr. Salvemini's group, its contribution in CIBP is unknown.
2. The behavioral characterization of Gpr160 role in CIBP is thoroughly performed

including AAV mediated knockdown, germline KO and use of two species.

However, following concerns dampen my enthusiasm somewhat.

Response: We thank reviewer greatly for the efforts and time put into the review of the manuscript, and the constructive comments and suggestions. All comments were contributed greatly on revising and improving our manuscript. We have studied these comments carefully and have made corresponding corrections that we hope will meet with your approval. The responses to the reviewer's comments are provided below.

Major:

1. The implication that only TRPV1 is mediating Gpr160 effects are not well supported. E.g. Fig 4 shows change in AP and current thresholds in the Gpr160 knockdown animals which suggests channels other than TRPV1 that underlie threshold setting are affected. Although the data in Fig 8 point towards increased TRPV1 function in CIBP DRG neurons that is dependent on presence of Gpr160, how presence or activation of Gpr160 leads to enhanced TRPV1 function remains unanswered.

Response: We thank the reviewer for raising this important issue. We appreciate the reviewer's insightful comments on the potential involvement of channels other than TRPV1 in mediating GPR160 effects. Indeed, the changes in action potential and current thresholds observed in the *Gpr160* knockdown animals suggest a broader spectrum of channels might be affected. While our data in Fig. 8 indicate increased TRPV1 function in BCP DRG neurons dependent on GPR160, the precise mechanisms by which GPR160 presence or activation enhances TRPV1 function remain to be elucidated. Taking into account the comments of all reviewers, we decided to remove the research on TRPV1 from this study. We acknowledge this limitation and plan to investigate this aspect in our next studies. Thank you for highlighting this critical point.

2. [REDACTED: Referee's comment and the author's response with unpublished data.]

[REDACTED: Referee's comment and the author's response with unpublished data.]

Figure for referee with unpublished data and its description has been removed upon request by the authors.

3. The data in Fig 8E-G suggests that capsaicin behavioral response is dependent on the presence of Gpr160. This is hard to explain given their own data that TRPV1 expression is not affected by absence of Gpr160. Are the authors suggesting that capsaicin does not bind to TRPV1 or TRPV1 is not present on plasma membrane in absence of Gpr160?

Response: Special thanks to you for your comments. [REDACTED: Referee's comment and the author's response with unpublished data.]. Recent research has revealed that Sp1, a transcription factor, functions as a cyclin-dependent kinase-5 (CDK5) activator and plays a pivotal role in inflammatory thermal hyperalgesia by impairing TRPV1 internalization(Adwan et al., 2015, Liu et al., 2019). In the context of BCP, Sp1 assumes a critical role as a transcriptional regulator of GPR160. Our conjecture suggests that the attenuated responsiveness to capsaicin observed in Gpr160 knockout (KO) mice may stem from heightened TRPV1 internalization. This intriguing result requires further investigation.

Figure for referee with unpublished data and its description has been removed upon request by the authors.

4. Gpr160 overexpression model could be utilized to identify pathways by which TRPV1 activity is enhanced to consolidate the hypothesis.

Response: We thank the reviewer for the valuable suggestion. Based on the previous suggestions of reviewer, in our next study, we plan to add TRPV1 plasmid and TRPV1 + GPR160 plasmid separately to HEK293T cells to observe the calcium response through calcium imaging experiments.

5. The concentration of capsaicin used in Fig 8 calcium imaging experiments is 500 micromolar. This is orders of magnitude above the typical concentrations of capsaicin used in these experiments by many other groups. Is this a typo?

Response: Thank you honestly for pointing out this mistake. The concentration of capsaicin used in the Fig. 8 calcium imaging experiments was indeed misstated, the actual concentration is 500 nanomoles.

Minor:

1. Fig 1: It would be useful to show separate marker images along with total neuronal

count before the merged image (N and O) so that the level of co-expression can be assessed better. Inclusion of a colocalization coefficient analysis would also be helpful (also applies to other IHC figures).

Response: We thank the reviewers for their constructive feedback. In response, we have provided separate images showing GPR160 with markers for neurons and glial cells before presenting the merged images in Fig. 1K-T. Additionally, we have quantified other IHC figures the co-localization and included the colocalization coefficient analysis to better assess the level of co-expression (Fig 6F and Fig 7H). Thank you for your valuable suggestions and corresponding changes has been added to the manuscript.

Fig 1K-T

Fig 6A-F

Fig 7G and H

2. Fig 3: It would help the reader if description of the groups (i.e. sham/sham+sh etc.) were listed together rather than on different panels.

Response: Thank you for your suggestions. We have revised the manuscript to list the descriptions of the groups together, rather than on different panels, to enhance clarity and ease of reading for the audience. We displayed the mechanical pain, thermal and cold allodynia of rats and mice separately, and matched them with schematic diagrams and flowcharts (Fig3, Fig4, and Fig EV2). Corresponding changes has been added to the manuscript.

Fig 3

Fig 4

Fig EV2

3. Fig 5: A and C: The western blots do not have the protein lanes well separated making quantification difficult.

Response: As the reviewer suggested, we have rerun the Western blot experiments to ensure better separation of the protein lanes (Fig 6A and C). This will facilitate more accurate quantification and statistical analysis. In this experiment, we separated nuclear proteins from cytoplasmic proteins, and small molecule proteins are prone to deformation and binding during electrophoresis. The statistics of this experiment were semi quantitatively analyzed using Image J software.

Fig 6A-D

Referee #2:

Authors Xu et al., explored the role of orphan GPCR GPR160 in bone cancer pain (BCP) modelled using multiple in vivo rodent models, specifically relating to hyperexcitability in small diameter neurons in the DRG. A large number of behavioral tests, along with transgenic mice, siRNA and AAV tools, and patch clamp showed a key role for Gpr160 in the development of hypersensitivity. DNA, RNA and protein variability and expression were assessed, to relate an increase in Gpr160 mRNA and protein in the DRG to increased binding of transcription factor Sp1 driven by histone modification (reduced H3K27me3 and increased H3K27ac). Increased GPR160 was subsequently associated with increased TRPV1 function in the DRG using calcium imaging and behavior. In this manuscript, multiple techniques are used to comprehensively characterize the role of H3K27me3/H3K27ac-SP1-GPR160 in the development of hypersensitivity in both mice and rats. The link to increased TRPV1 function is found to be weaker, despite this, the manuscript comprehensively characterizes an interesting and understudied pathway. However, a number of major and minor points should be addressed:

Response: We thank the reviewer for the positive comments. Moving forward, we are committed to advancing this project with the aim of yielding further promising outcomes.

1. While concordant results from multiple models are compelling (despite the cancer-cell and tibia/femur bone variation), the sheer volume of data presented makes it difficult to easily distinguish what the source of the data is. An effort should be made to differentiate more clearly both in the methods and in the results. Adding timelines for the use of different techniques in each model, as well as for cell-culture experiments would aid the legibility. Additionally, reorganization of methods is recommended to reflect whether a technique was used for mice or rats, or even primary cultures.

Response: We appreciate the reviewer's valuable suggestion. In response, we have now presented the behavioral assays for rats and mice separately in the **Methods and Results** sections. Furthermore, we have distinguished between rat and mouse results in the Results section by utilizing pattern plots and timelines (Fig3, Fig4, and Fig EV2). This reorganization enhances the clarity and readability of the data, facilitating the distinction between the different sources. Additionally, we have included a detailed description of the cell culture process in the Methods section, outlining the preparation of Walker 256 and LLC cells used to establish rat and mouse models of

BCP, respectively. We are grateful for your constructive feedback. Corresponding modifications have been made in the manuscript and highlighted in red.

Fig 3

Fig 4

Fig EV2

2. The genome-wide mRNA expression profiling analysis in figure 1D, E, F is unclear, what is the source of this data? Adding details of the design of this experiment (n's, methods, statistical analysis etc) would aid in the interpretation of the results.

Response: Thank you for the advice. We have added a detailed description of the mRNA microarray experiment in the Methods section. Additionally, the mRNA microarray data has been deposited in the NCBI GEO database under accession code GSE270839 (<https://www.ncbi.nlm.nih.gov/geo/query/acc.cgi?acc=GSE270839>).

Corresponding changes has been added to the manuscript.

3. Briefly, experiments with an additional rat model (MADB-breast cancer cells) are presented and sex-differentiation is addressed in supp F2 with this model. Addition of stats of sex-matching/differentiation in the other models would be valuable. In contrast, the addition of results from the MADB-model are not found to add significant value to the story at this time, perhaps exclusion to prioritize clarification of other data would be beneficial.

Response: We appreciate your advice. Pain is known to exhibit sexual dimorphism,

and our objective was to investigate potential sexual dimorphism in pain mechanisms related to GPR160. Compared to Walker256 cells, MADB cells demonstrated a higher success rate in establishing BCP models in both female and male subjects. Consequently, we selected MADB cells for the creation of BCP models to explore sexual dimorphism in the mechanism of pain. However, due to the limited availability of relevant research, we agree with the reviewer's suggestion to remove this section from the study. The research on sexual dimorphism in the mechanism of pain will be conducted in the next step.

4. Experiments with a PC12 cell line (co-immunoprecipitation, luciferase reporter assay) are described, however, the source of cell line is not mentioned in the methods.

Response: Thank you for pointing this out. We have added the source of the PC12 cell line to the Methods section. The PC12 cells were purchased from the cell bank of the Chinese Academy of Sciences (Shanghai, China). We appreciate your attention to this detail. Corresponding modifications have been made in the manuscript and highlighted in red.

5. The tests used to collect behavioral results, particularly presented in figure 3 and figure 8, are unclear in the current format, especially considering the large volume of behavioral tests described in the methods. A redesign of the figure is recommended to aid legibility.

Response: Thank you for your valuable advice. In response, we have redesigned the presentation of the behavioral data to enhance readability. The behavioral results for rats and mice are now presented separately, using schematics and flowcharts to clarify the distinctions (See Fig3, Fig4, and Fig EV2 in the response question 1 above). Additionally, we have delineated the differences between the rat and mouse models in the Methods section. Corresponding modifications have been made in the manuscript and highlighted in red.

6. In the presentation of calcium imaging results in figure 8, there appears to be a discrepancy between the KCl concentration indicated in the legend (40mM) and the methods (50mM). Additionally, a 500uM concentration of capsaicin sounds disproportionately high, perhaps this is an error? Additionally, in Figure 8E-G the capsaicin injection is given as 10ug, and the injection volume or timeline of experiment are not characterized in the methods.

Response: We apologize for the inconsistencies. The correct concentration of KCl

used in the calcium imaging experiments was 50 mM, as stated in the methods, and we will update the figure legend to reflect this. The concentration of capsaicin was indeed misstated; the correct concentration was 500 nM. Additionally, mice were injected with 10 µg capsaicin (10 µl) in the left footpad using a Hamilton syringe. Detect mechanical and thermal pain in mice within 120 minutes after intraplantar injection. For the spontaneous pain test, mice were evaluated by quantifying the duration of paw licking (in seconds) on the affected limb every minute over a 5-minute period following the intraplantar injection of capsaicin. Thank you honestly for pointing out this mistake.

7. In figure 4 the effect of BCP on excitability of small-diameter neurons is assessed with patch, Gpr160 KO is associated with decreased A_p , however, n-number is unclear, are these neurons from 1 animal? Also, a change is noted in the rheobase between cells from Wt/KO-BCP mice and AAV-injected (for over-expression of Gpr160) mice, perhaps this can be addressed in the results?

Response: Comments appreciated. Following your suggestion, we have added the number of n's to the figure notes, indicating that the neurons are from 3-4 animals. However, there is a change in the rheobase between cells from Wt/KO-BCP mice and AAV-injected mice, which may be due to many other factors. Different batches of primary cultured neurons, different environmental temperatures for cell detection, etc. may affect the excitability of the neurons, thereby altering their response to electrical stimulation and leading to changes in rheobase (Viatchenko-Karpinski and Gu, 2018). Thank you for highlighting these important points. Corresponding changes has been added to the manuscript.

8. While the mechanism of Gpr160 is well characterized, the link to TRPV1 function is less convincing at this time. Co-expression of Gpr160 and TRPV1 in small-diameter neurons, as well as BCP affecting TRPV1 function are expected results. Lowered capsaicin response in Gpr160 KO cells and mice may be due to uncharacterized interactions, rather than direct effect of Gpr160 on TRPV1. Perhaps the title/introduction could be changed to refrain from predicting a direct correlation to TRPV1 function.

Response: We appreciate your suggestion. In this study, we obtained numerous results, including immunofluorescence, immunoprecipitation, and calcium imaging, which collectively suggest that GPR160 may influence TRPV1 function. However, we lack direct evidence, such as data demonstrating the direct interaction between membrane proteins and ion channels. Considering these factors, we concur with the reviewer's

comments and have decided to temporarily exclude the content related to TRPV1 from the manuscript, pending further research.

Referee #3:

"Histone modification promotes GPR160 contributes to bone cancer pain via regulating TRPV1 in rodents " reports novel results derived from insightful experiments with a topic of pathogenesis of bone cancer pain. - One could have said "... with a FOCUS on pathogenesis of bone cancer pain." But that is exactly the central weakness of the study, namely it is not strong on FOCUS.

On the other hand, the paper, in many of its sections, shows rigor and depth so that data do indeed support the authors' claims.

As alluded in the opener, my most important issue with this paper is that it represents a piece of work - a lot of hard work - that hurts itself by being overboarding, in a way overloading the plate and thus detracting from its value.

If we start with the title: "Histone modification promotes GPR160 contributes to bone cancer pain via regulating TRPV1 in rodents" - did the authors mean "Histone modification enhances GPR160 expression and function as an important contributor to bone cancer pain in rodents" ? And here is my point: this could be a suitable title, but then "Histone modification enhances GPR160 expression and function as an important contributor to bone cancer pain in rodents, with suggested co-contribution of upregulated TRPV1 function" clearly is odd and shows the dilemma of having generated an enormous amount of data and trying to force them under one headline.

Having read the paper carefully and repeatedly - which also made clear, more and more with every re-read, that the paper does not read well, likely driven by the overabundance of what authors aim to report: My take-home is that core elements of the study are the regulated expression of GPR160 in bone cancer pain in rodents, epigenetic mechanisms of Gpr160 gene regulation and in-vivo role of GPR160, the latter being studied using gene knockdown and knockout and also - it is laudable and bespeaking of the authors' diligence - gain-of-function studies.

It is my recommendation that the study should wrap these important findings up and explore the related TRPV1 mechanisms separately.

Demonstrated altered electrical excitability of DRG neurons is rather linked directly

to endogenous excitability machinery (voltage gated channels, K⁺ channels) than to TRP channels. Perhaps the authors can explore more of the mechanistic link between GPR160 and TRPV1, such as involvement of cell-autonomous signaling mechanisms involving phospholipids, lipases, kinases etc. and structural determinants in the GPR and the TRP channel. Also, whether this is a DRG mechanism that is coupled to the previously demonstrated co-function TRPV1-ANO1 (thus linking calcium influx with chloride eflux), which powerfully amplifies noci-transduction in DRG neurons.

Beyond careful proofing and editing for proper use of English the authors need to shorten their Discussion to enhance readability.

However, what needs to be included in the revised Discussion, guiding perhaps future experimentation, are references to known bone cancer pain mechanisms in clinics, and also more recent bone cancer pain mechanisms that are more directly coupled to mechanosensory mechanisms.

Re the latter, TRPV1 (and GPR160) are not strongly supported as involved in mechanical sensing/ mechanotransduction.

The clinical evidence is clear that NGF signaling and mu-opioid receptor signaling are critically involved in human bone cancer pain. - What could be the linke to the authors' new findings?

More recent preclinical studies suggest involvement of intrinsically mechanically sensitive signaling mechanisms in bone cancer pain, these should be discussed.

Response: Thank you for your insightful comments and recommendations regarding our manuscript. We have carefully reviewed your suggestions and made the following revisions:

We agree with your observation that the original manuscript lacked a clear focus. We have now restructured the paper to concentrate primarily on the epigenetic regulation of GPR160 in BCP. The involvement of TRPV1 has been addressed, but we have separated it from the core narrative, with plans to explore this mechanism further in future studies.

Based on your suggestion, we have revised the title to better reflect the central theme of the study. The new title is: " Histone modification enhances GPR160 expression and function as an important contributor to bone cancer pain in rodents". This title

more accurately captures the primary findings without overloading the scope.

We have shortened and refined the Discussion to improve readability and focus. We also incorporated references to established BCP mechanisms in clinical settings and discussed recent findings related to mechanosensory mechanisms, as you recommended. Additionally, we clarified the potential mechanistic link between GPR160 and TRPV1, though we acknowledge that this aspect requires further investigation in future work. We have carefully proofread the manuscript to correct any language issues and enhance clarity.

In clinical practice, the administration of antibodies that inhibit nerve growth factor (NGF) function, as well as small molecule inhibitors targeting NGF receptors, has been shown to provide significant pain relief. However, these treatments are associated with a risk of accelerating the progression of osteoarthritis in patients (Wise et al., 2021). Mu-opioid receptors are known for their potent analgesic effects; however, their prolonged clinical use is restricted due to the potential for significant adverse effects (Srivastava et al., 2020). Currently, no connection has been established between orphan G protein-coupled receptors (oGPCRs), including GPR160, and the NGF or μ -opioid receptor signaling pathways. Despite this, oGPCRs, including GPR160, play a significant role in BCP. Our study indicates that GPR160 is involved in BCP; however, the clinical efficacy and safety of anti-GPR160 antibodies and small molecule inhibitors have yet to be investigated. Future research should address these aspects to establish GPR160 as a viable target for adjuvant therapy or potential replacement of existing NGF and μ -opioid receptor treatments.

Our study results indicate that GPR160 affects both the mechanical pain, cold allodynia and thermal hyperalgesia in BCP, which may be regulated by different mechanisms. Recent preclinical studies highlight the importance of mechanically sensitive signaling pathways, such as STING and PD-1/PD-L1 (Wang et al., 2021, Liu et al., 2020), in the development and maintenance of BCP. These pathways, along with other mechanisms like VEGF-A/VEGFR2-driven central sensitization and the TREK-1 potassium channel (Hu et al., 2019, Delanne-Cuménal et al., 2024), provide new insights and potential therapeutic targets for managing BCP. Cold and thermic sensitive signaling mechanisms play an important role in pain. The transient receptor potential channels (TRP), also known as thermoTRP channels, intervene in the perception of hot and cold external stimuli. These channels, and especially TRPV1, TRPA1 and TRPM8, have been subjected to profound investigation because of their role as thermosensors and also because of their implication in acute and chronic pain (Cabañero et al., 2022). Recent research has revealed that Sp1 functions as a

cyclin-dependent kinase-5 (CDK5) activator and plays a pivotal role in inflammatory thermal hyperalgesia by impairing TRPV1 internalization (Adwan et al., 2015, Liu et al., 2019). It suggests that GPR160 may influence TRPV1 internalization. Therefore, it is essential to explore the mechanically, cold and thermic sensitive signaling mechanisms of GPR160 in future studies.

We have incorporated recent preclinical studies that suggest the involvement of mechanically sensitive signaling mechanisms in bone cancer pain. This discussion highlights the potential relevance of our findings to these emerging areas of research.

Finally, three specific details appear worthy of comment:

1) The authors state "Given that the initial pathological alterations in DRG serve as the primary instigators of chronic pain, ..."

The fact is that there are virtually certainly other primary instigators of chronic pain OUTSIDE the DRG.

Response: Thank you for your valuable feedback. We acknowledge your point regarding the statement about the initial pathological alterations in the DRG serving as primary instigators of chronic pain. We agree that chronic pain can indeed originate from multiple sources outside the DRG, including peripheral tissues, spinal cord, and higher central nervous system structures. We have revised the manuscript to make it more descriptive" Given that the initial pathological alterations in the DRG are one of the important elements initiating chronic pain". Corresponding changes has been added to the manuscript.

2) Fig. 1 - antibody validation needs to be conducted with Gpr160 knockout mouse tissue, such a finding conveys higher rigor than blocking of antibody binding with immunizing peptide.

Response: Thank you for the suggestion. In response, we have used GPR160 knockout mouse to validate the specificity of the antibody (Fig S1D). This method conveys higher rigor than blocking antibody binding with an immunizing peptide. We appreciate your input to enhance the quality of our research.

Fig S1D

3) Define the AAV2/9 capsid better - AAV9 typically has been used for targeting neural substrate, which has now been replaced with the PHP.S capsid (a AAV9 derivative) when targeting DRG neurons.

Response: We are very grateful to the reviewer for their suggestion regarding the AAV capsid specifically targeting DRG neurons. In this study, we used in situ injection of AAV with DRG, which increased the difficulty of manipulation, and we experimentally verified the effect of its transfection. Successfully transfected into DRG neurons, but not in the spinal dorsal horn. For future experiments, we will use AAV vectors more specifically targeting DRG neurons, such as the PHP.S capsid, to ensure the effectiveness and precision of AAV transfection. Thank you for your valuable input. Corresponding changes has been added to the manuscript.

Fig EV3A-C

References

- ADWAN, L., SUBAIEA, G. M., BASHA, R. & ZAWIA, N. H. 2015. Tolfenamic acid reduces tau and CDK5 levels: implications for dementia and tauopathies. *J Neurochem*, 133, 266-72.
- CABAÑERO, D., VILLALBA-RIQUELME, E., FERNÁNDEZ-BALLESTER, G., FERNÁNDEZ-CARVAJAL, A. & FERRER-MONTIEL, A. 2022. ThermoTRP channels

- in pain sexual dimorphism: new insights for drug intervention. *Pharmacol Ther*, 240, 108297.
- DELANNE-CUMÉNAL, M., LAMOINE, S., MELEINE, M., AISSOUNI, Y., PRIVAL, L., FERAYROLLES, M., BARBIER, J., CERCY, C., BOUDIEU, L., SCHOPP, J., LAZDUNSKI, M., ESCHALIER, A., LOLIGNIER, S. & BUSSEROLLES, J. 2024. The TREK-1 potassium channel is involved in both the analgesic and anti-proliferative effects of riluzole in bone cancer pain. *Biomed Pharmacother*, 176, 116887.
- HU, X. M., YANG, W., DU, L. X., CUI, W. Q., MI, W. L., MAO-YING, Q. L., CHU, Y. X. & WANG, Y. Q. 2019. Vascular Endothelial Growth Factor A Signaling Promotes Spinal Central Sensitization and Pain-related Behaviors in Female Rats with Bone Cancer. *Anesthesiology*, 131, 1125-1147.
- LIU, B. L., CAO, Q. L., ZHAO, X., LIU, H. Z. & ZHANG, Y. Q. 2020. Inhibition of TRPV1 by SHP-1 in nociceptive primary sensory neurons is critical in PD-L1 analgesia. *JCI Insight*, 5.
- LIU, J., DU, J. & WANG, Y. 2019. CDK5 inhibits the clathrin-dependent internalization of TRPV1 by phosphorylating the clathrin adaptor protein AP2 μ 2. *Sci Signal*, 12.
- SRIVASTAVA, A. B., MARIANI, J. J. & LEVIN, F. R. 2020. New directions in the treatment of opioid withdrawal. *Lancet*, 395, 1938-1948.
- VIATCHENKO-KARPINSKI, V. & GU, J. G. 2018. Effects of cooling temperatures and low pH on membrane properties and voltage-dependent currents of rat nociceptive-like trigeminal ganglion neurons. *Mol Pain*, 14, 1744806918814350.
- WANG, K., DONNELLY, C. R., JIANG, C., LIAO, Y., LUO, X., TAO, X., BANG, S., MCGINNIS, A., LEE, M., HILTON, M. J. & JI, R. R. 2021. STING suppresses bone cancer pain via immune and neuronal modulation. *Nat Commun*, 12, 4558.
- WISE, B. L., SEIDEL, M. F. & LANE, N. E. 2021. The evolution of nerve growth factor inhibition in clinical medicine. *Nat Rev Rheumatol*, 17, 34-46.
- YIN, Y., LE, S. C., HSU, A. L., BORGNIA, M. J., YANG, H. & LEE, S. Y. 2019. Structural basis of cooling agent and lipid sensing by the cold-activated TRPM8 channel. *Science*, 363.
- YIN, Y., ZHANG, F., FENG, S., BUTAY, K. J., BORGNIA, M. J., IM, W. & LEE, S. Y. 2022. Activation mechanism of the mouse cold-sensing TRPM8 channel by cooling agonist and PIP(2). *Science*, 378, eadd1268.

Dear Prof. Ni

Thank you for the submission of your revised manuscript to EMBO reports. We have now received the full set of referee reports that is copied below.

As you will see, both referees are very positive about the study and support its publication.

Browsing through the manuscript myself, I noticed a few editorial things that we need before we can proceed with the official acceptance of your study.

- Please note that all corresponding authors need to provide an ORCID ID. This information is currently missing for Prof. Yao and Prof. Lin. Please find instructions on how to link your ORCID ID to your account in our manuscript tracking system in our Author guidelines

(<<https://www.embopress.org/page/journal/14693178/authorguide#authorshipguidelines>>)

- Please provide up to 5 keywords.

- The funding information in the manuscript and in our online manuscript tracking system must be congruent. Please enter the following funding information in the online tracking system: the China Postdoctoral Science Foundation (2022M712310), the Medical and Health General Research Program of Zhejiang Province (2021RC130), the National Clinical Key Specialty Construction Project-Oncology department (2023-GJZK-001), the Zhejiang Provincial Traditional Chinese Medical Innovation Team of China under Grant No. 2022-19, the Key Medical Subjects Established by Zhejiang Province and Jiaxing City Jointly Pain Medicine and Oncology (2019-ssstyx, 2023-SSGJ-001), the Clinical Key Specialty of Zhejiang Province Anesthesiology (2023-ZJZK001), the Construction Project of Key Laboratory of Nerve and Pain Medicine in Jiaxing City, Scientific research project of Anhui Provincial Health Commission (AHWJ2023A30069), the Clinical Key Specialty of Anhui Province Anesthesiology (2022-AH-105) and Bengbu Science and Technology Projects (20220115)

- Please update the 'Conflict of interest' paragraph to our new 'Disclosure and competing interests statement'. For more information see

<https://www.embopress.org/page/journal/14693178/authorguide#conflictsofinterest>

- Regarding the Author Contributions, we now use CRediT to specify the contributions of each author in the journal submission system. Therefore, please remove the Author Contributions from the manuscript file and make sure that the author contributions in our online manuscript tracking system are correct and up-to-date. The information you specified in the system will be automatically retrieved and typeset into the article. You can enter additional information in the free text box provided, if you wish.

- Figure callouts:

a) Please add callouts for Fig 5I, 5L-N, Appendix Fig. S4, and Appendix Table S2-S3 in the text.

b) There are callouts for Fig. 8 A-G but Fig. 8 does not contain any panels. This needs to be rectified.

c) There are callouts for Supplementary Fig 1N and O and Supplementary Fig 2K-L but no such figures were uploaded. This needs to be rectified.

- Please include the EV figure legends in the manuscript file after main figure legends.

- Figures S1-S3 with their legends and the Appendix tables should be compiled in one Appendix PDF with a table of content listing the items with corresponding page numbers on the title page. Table S1 and S2 could be included in the Reagent and Tools table instead of the Appendix (see next point).

- Since July 1st, all our manuscript should contain a Reagents and Tools Table (listing key reagents, experimental models, software and relevant equipment and including their sources and relevant identifiers).

Please download and fill our Reagents and Tools Table template (.docx), which you can find in our author guidelines:

When submitting your revised manuscript, please do not include the Reagents and Tools Table in the Methods section of the manuscript but upload it as a separate file choosing the file type "Reagent Table". Our typesetters will then include the table in the methods section for you.

- Author Checklist

a) You answered the item "Animal observed in or captured from the field". Since you exclusively used laboratory rat and mouse strains, as far as I could see, you should choose "Not applicable" instead of YES here.

b) Authority granting ethics approval: Please include the reference number for the approval in the Methods section as well.

- Data availability statement should only refer to data deposited in public repositories. Therefore, please remove the following statements from this paragraph: "The original data in this study are provided in the Source Data file."

- Source data: Please export the .pzf files as .csv file. You can include both formats in the source data but the '.csv' files are 'readable' for many softwares and would thus ease the reuse by others.

- Our production/data editors have asked you to clarify several points in the figure legends (see below). Please incorporate these changes in the manuscript and return the revised file with tracked changes with your final manuscript submission:

In general, please ensure that the scale bars are thick enough to be visible at final size. The scale bars in Figure 1 K-S, e.g., appear rather thin. Moreover, statistics analysis should only be done if the sample size is at least 3 biological replicates.

A) Statistical test information. Only p-values that are actually shown in the figure panel(s) should (and must) be defined in the legends, all others should be removed from (or added to) the legend. Moreover, we ask for the specification of exact p-values:

1. Please note that the figure EV 1n; EV 5a-g does not contain any p values, kindly rectify the statistical test related information in the figure legend appropriately.

2. Please define the annotated p values **/* as well as provide the exact p-values for the same in the legend of figure 6f; 7h; EV 4c, as appropriate.

3. Please note that the exact p values are not provided in the legends of figures 1d-f; 3c-g, i-k; 4c-h; 5g, j-k, n; 6g, i-j; 7b-f, k; EV 1a-c, e-h, j-k; EV 2b, d-f, h-j; EV 3e, g; EV 4d-f, i-l, n-o; EV 5i-j, l, n.

4. Please indicate the statistical test used for data analysis in the legends of figures 1b; 6f; 7h.

5. Please note that in figures 6b, d; 7j; there is a mismatch between the annotated p values in the figure legend and the annotated p values in the figure file that should be corrected.

B) Replicates and error bars:

6. Please note that information related to n is missing in the legends of figures 1b, o, t; 5i-l; 6f-g; 7b-d, h.

7. Although 'n' is provided, please describe the nature of entity for 'n' in the legends of figures 2j, o, t.

8. Please note that the error bars are not defined in the legends of figures 1d-f; 2e, j, o, t; 3c-k; 4c-h; 5g, i-l, n; 6b, d, f-g, i-j; 7b-f, h, j-k; EV 1a-c, e-h, j-k, n; EV 2b, d-j; EV 3e, g; EV 4c-g, i-l, n-o; EV 5a-g, i-j, l, n.

C) Data presentation:

9. Please note that scale bar and its definition are missing for figures 2a-c, f-h, k-m, p-r.

10. Please note that the red arrow is not defined in the legend of figure EV 2a, c, EV 3d, f; This needs to be rectified.

11. Please note that the white arrows are not defined in the legends of figures 2d, i, n, s; 7g, i. This needs to be rectified.

- The section order should be corrected: title page with complete author information, abstract, keywords, introduction, results, discussion, methods, data availability section, acknowledgements, disclosure and competing interests statement, references, main figure legends, tables, expanded figure legends.

- As a standard procedure, we edit the title and abstract of manuscripts to make them more accessible to a general readership. Please find the edited versions below my signature and let me know if you do NOT agree with any of the changes.

- Finally, EMBO Reports papers are accompanied online by

A) a short (1-2 sentences) summary of the findings and their significance,

B) 2-3 bullet points highlighting key results and

C) a schematic summary figure that provides a sketch of the major findings (not a data image).

Please provide the summary figure as a separate file in PNG or JPG format at a size of 550x300-600 pixels (width x height).

Please note that the size is rather small and that text needs to be readable at the final size. Please send us this information along with the revised manuscript.

- On a different note, I would like to alert you that EMBO Press offers a new format for a video-synopsis of work published with us, which essentially is a short, author-generated film explaining the core findings in hand drawings, and, as we believe, can be very useful to increase visibility of the work. This has proven to offer a nice opportunity for exposure i.p. for the first author(s) of the study. Please see the following link for representative examples and their integration into the article web page:

<https://www.embopress.org/doi/full/10.15252/emj.2019103932>

With kind regards,

Referee #2:

During the initial review several issues were identified - these are all found to be addressed and accounted for in this submission. I have no further concerns.

Referee #3:

The authors have satisfactorily addressed all my points of critique.

Histone modifications and Sp1 promote GPR160 expression and bone cancer pain in rodents

Bone cancer pain (BCP) affects approximately 70% of patients in advanced stages, primarily due to bone metastasis, presenting a substantial therapeutic challenge. Here, we profile orphan G protein-coupled receptors in the dorsal root ganglia (DRG) following tumor infiltration, and observe a notable increase in GPR160 expression. Elevated Gpr160 mRNA and protein levels persist from postoperative day 6 for over 18 days in the affected DRG, predominantly in small-diameter C-fiber type neurons specific to the tibia. Targeted interventions, including DRG microinjection of siRNA or AAV delivery, mitigate mechanical allodynia, cold, and heat hyperalgesia induced by the tumor. Tumor infiltration increases DRG neuron excitability in wild-type mice, but not in Gpr160 gene knockout mice. Tumor infiltration results in reduced H3K27me3 and increased H3K27ac modifications, enhanced binding of the transcription activator Sp1 to the Gpr160 gene promoter region, and induction of GPR160 expression. Modulating histone-modifying enzymes effectively alleviated pain behavior. Our study delineates a novel mechanism wherein elevated Sp1 levels facilitate Gpr160 gene transcription in nociceptive DRG neurons during BCP in rodents.

All editorial and formatting issues were resolved by the authors.

Prof. Huadong Ni
The Affiliated Hospital of Jiaxing University
China

Dear Prof. Ni,

Thank you for implementing the final minor editorial changes. I am very pleased to accept your manuscript for publication in the next available issue of EMBO reports. Thank you for your contribution to our journal.

Yours sincerely,
